# Competitive exclusion during co-infection as a strategy to prevent the spread of a virus: A computational perspective

Safar Vafadar[1◉], Maryam Shahdoust[2◉], Ata Kalirad[2◉], Pooya Zakeri[3,4◉], Mehdi Sadeghi[2,5]*

1 Laboratory of Biological Complex Systems and Bioinformatics (CBB), Institute of Biochemistry and Biophysics, University of Tehran, Tehran, Iran, 2 School of Biological Science, Institute for Research in Fundamental Sciences (IPM), Tehran, Iran, 3 Centre for Brain and Disease Research, Flanders Institute for Biotechnology (VIB), Leuven, Belgium, 4 Department of Neurosciences and Leuven Brain Institute, KU Leuven, Leuven, Belgium, 5 National Institute of Genetic Engineering and Biotechnology (NIGEB), Tehran, Iran

◉ These authors contributed equally to this work.
* sadeghi@nigeb.ac.ir

**Data Availability Statement:** The program written simulate the spread of the competition between

## Abstract

Inspired by the competition exclusion principle, this work aims at providing a computational framework to explore the theoretical feasibility of viral co-infection as a possible strategy to reduce the spread of a fatal strain in a population. We propose a stochastic-based model—called Co-Wish—to understand how competition between two viruses over a shared niche can affect the spread of each virus in infected tissue. To demonstrate the co-infection of two viruses, we first simulate the characteristics of two virus growth processes separately. Then, we examine their interactions until one can dominate the other. We use Co-Wish to explore how the model varies as the parameters of each virus growth process change when two viruses infect the host simultaneously. We will also investigate the effect of the delayed initiation of each infection. Moreover, Co-Wish not only examines the co-infection at the cell level but also includes the innate immune response during viral infection. The results highlight that the waiting times in the five stages of the viral infection of a cell in the model—namely attachment, penetration, eclipse, replication, and release—play an essential role in the competition between the two viruses. While it could prove challenging to fully understand the therapeutic potentials of viral co-infection, we discuss that our theoretical framework hints at an intriguing research direction in applying co-infection dynamics in controlling any viral outbreak's speed.

## Introduction

The "competitive exclusion principle" (CEP), also—perhaps erroneously—known as "Gause's law" [1], is the consequence of natural selection operating on non-interbreeding populations that occupy the same ecological niche. As Darwin put it, "the competition will

viruses is available at https://github.com/
safarvafadar/virusinfection.

**Funding:** The authors received no specific funding
for this work.

**Competing interests:** The authors have declared
that no competing interests exist.

generally be most severe [. . .] between the forms which are most like each other in all respects" [2] (p.320). In the simplest reading, this principle implies that, in the competition between two sympatric non-interbreeding populations over the same ecological niche, one will displace the other.

The crux of this work is to couple the CEP with the life cycle of viruses. To model how two types of viruses, *B* and *M*, co-infect a tissue, we have to make a few assumptions:

1. *Different viruses can have eclipse phases of varying lengths.* The period between the initial infection and the first detectable viremia is known as the eclipse period—from the moment a virus enters the cell until it starts assembling its progenies and subsequences burst out of the cell. The duration of this period varies among different virus strains and affects the kinetic of infection (e.g., the eclipse period of SHIV lasts around a day [3] while it lasts between 7 to 8 hours in SARS-CoV [4]).

2. *Different viruses have different burst sizes, i.e., different per-cell virion particles.* This assumption is reasonable, but even for SARS-CoV-2, the estimated burst size of $10^3$ is simply based on MHV-2 data (e.g., [5]). The variation in particle-to-PFU among animal viruses [6] can be used to deduce the veracity of this assumption, but more direct data on SARS-CoV-2 is needed.

3. *The CEP applies when two virus strains compete to infect a cell.* In a spatially heterogeneous environment, different populations tend to partition the environment into non-overlapping micro-environments; for one of the most famous experimental demonstrations of such a situation see [7]. However, the displacement of one of the competitors by another is inescapable when the competitors cannot adapt or construct new niches in the environment in a reasonable timescale. The notion of CEP is undisputed when it comes to animals and bacteria trying to occupy the same ecological niche, but its application to viral confection is not as unequivocal as one would imagine.

In addition, different viruses can have different incubation times. For example, the median incubation period—i.e., the period between the onset of infection and the appearance of symptoms—for SARS-CoV-2 is estimated to be 5.1 days [8] (although such estimates should be taken with a grain of salt, e.g., [9]), whereas the incubation period for the common cold is around 1–3 days, and it could even be as long a few months to few years (e.g., Rabies and AIDS) [6]. Here, we do not consider any designation to simulate the different incubation times of two imaginary viruses.

There are two major lines of investigations that can illuminate the applicability of the CEP to viral infections in general. These include (i) the experimental evidence on the negative interactions between viruses, and (ii) the computational models of co-infection. In relation to the first line of research, a recent paper [10] employed a population-level approach, based on 44230 cases over 9 years, to track the epidemiological interactions between 11 strains of respiratory viruses, including influenza A and B, rhinoviruses, and three human coronaviruses (229E, NL63, HKU1). They inferred negative interaction between Rhinoviruses (A–C) and Influenza A virus—at both population and host levels—and negative interaction between Influenza A virus and Influenza B virus—at the population level. In addition, the three human coronaviruses showed negative interactions with Rhinoviruses (A–C), human Respiroviruses 1 and 4 at the population level.

While the aforementioned approach is a rarity in studying an organism, models that describe the co-infection of two viruses in a single host have gained growing attention over the past few years (e.g. [11, 12]). Gonzalez et al. [11] have used a mouse model to show how infection from rhinovirus strain 1B—before exposure to influenza A virus—reduces the

severity of influenza in mice. They observed the same effect on the outcome of influenza when initially exposed to the hepatitis virus strain-1. Also recently, a more sophisticated experimental method using single-cell RNA localization has described how the entry of an influenza A virus restricts the replication of a second viral genome in the same cell [13]. This growing body of research highlights the prevalence of interaction between viruses in the same host; though many aspects of co-infection dynamics have yet to be experimentally elucidated.

The computational models of co-infection have primarily been concerned with the co-infection of a cell by multiple copies of the same virus, where these copies compete over target cells and resources (e.g., [14]). Here, we propose a stochastic-based model to explore the possible effects of co-infection by two species of virus. Although deterministic mathematical methods based on ordinary differential equations (ODE) are typical approaches to study the dynamics of viral infection, they often ignore the stochastic nature of the viral growth process. It has been shown that they can sometimes lead to inaccurate results because they mainly rely on the average reactions of viral infections [15–19]). To simulate the stochastic multi-stage nature of the viral growth process occurring in viral co-infection, Gillespie algorithm, also known as the stochastic simulation algorithm, is usually considered as a trustworthy alternative. In particular, it supports simulating both discrete and stochastic behaviors of each stage of viral growth. The algorithm explicitly simulates the behavior of each virus in each stage of its growth process. Finally, it can provide a flexible framework to study the stochastic effects induced during the evolution of the infection. One characteristic of our simulation model is that several Gillespie algorithms run in infected cells simultaneously. This characteristic can represent what we presume of viral infection in one tissue. Gillespie's algorithms are indeed independent of their features. To illustrate the co-infection of two viruses using Gillespie's algorithm [20], we look at the dynamics of co-infection by introducing a benign and a malignant virus strains into a lattice of virtual cells. We investigate how the delay between the infection of the tissue by each strain, henceforth referred to as "co-infection delay" ($\mathcal{D}$), affects the competition between the two strains.

The proposed model—called Co-Wish—offers a more in-depth study of the interaction between different virus strains competing in a shared or overlapping niche. The model could eventually be used to design strategies to induce immunity in the population against a deadly virus by introducing a competing but less-deadly strain into the population. The results suggest that when one virus has a slight advantage—e.g., a virus with a shorter infection cycle—over another, the one with the advantage will eventually dominate the other.

## Results

To model the viral infection of tissue, we applied Gillespie's stochastic algorithm [20] to simulate the spread of infection over the lattice of virtual cells. In our model—Co-Wish—viral infection is a five-stage process including attachment, penetration, eclipse, replication, and release. We treat each step of infection as a probabilistic process (Fig 1).

The viral growth is simulated by considering the duration of each stage (waiting time) and the probability of transition from one stage into the next (transition probability). The waiting times follow either an exponential, $Exp(\lambda)$, or a Weibull distribution, $W(\lambda, \omega)$. All transition probabilities are generated from Beta distribution. The parameter values of the Beta distribution depend on the chosen distribution for the waiting time of the current stage. In the Methods section, we provide a detailed explanation of each of these steps in our model. In addition, we simulate the effect of the immune system on the dynamics of viral infection and how the interaction between virus strains affects their spread.

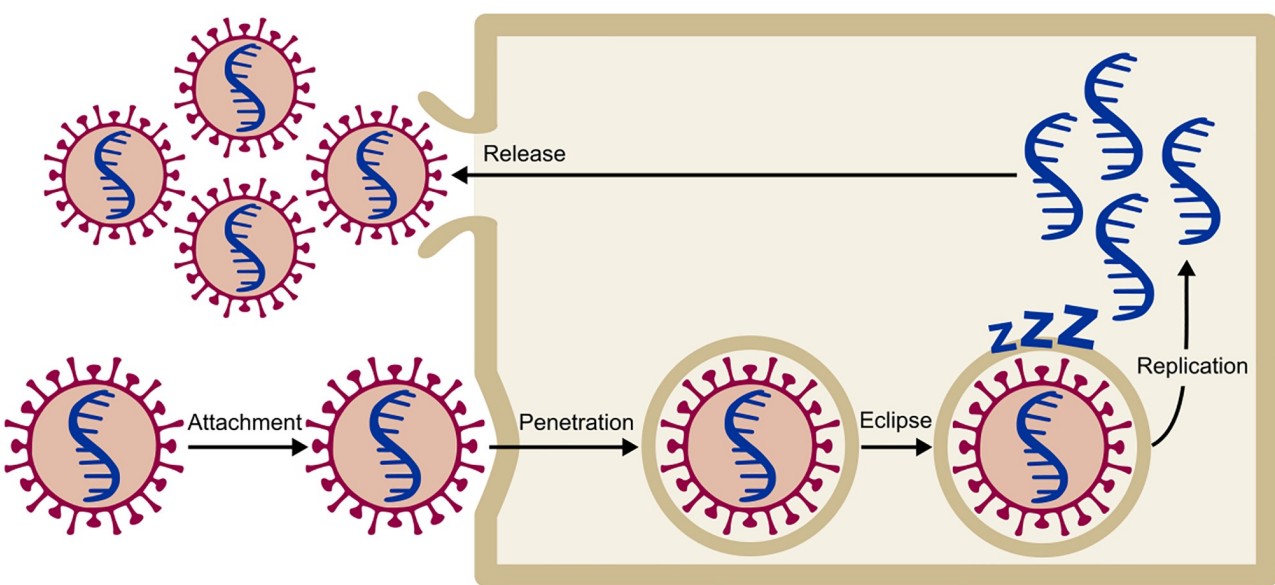

**Fig 1. The four stages of the viral infection of a cell in the model.** The virus can move from stage $i$ to stage $j$ according to the transition probability $P_{ij}$.

## The dynamics of single strain infection

Before investigating the dynamics of co-infection in our model, we assessed the effects of different parameters of our model when a single strain infects the tissue (Fig 2). We ran the model in two different conditions. At first, the viral infection was simulated by setting the parameters of the models as default values(all the parameters were set as one.). Then, we changed the parameters due to shortening the waiting times.

The results indicate that the shorter waiting times for different stages of the virus growth results in more viruses produced per infected cell and an increase in the number of infected cells. The impact of decreasing the waiting time on the production of new virions is more perceptible in the eclipse stage (Fig 2—Eclipse to Replication). Decreasing the waiting time at the eclipse stage increases the number of virions. Furthermore, it seems that the duration of the attachment stage directly affects the numbers of virions produced during the infection cycle (Fig 2—Attachment to Penetration)—perhaps because of a faster transition into the penetration stage allows the virus to replicate more virions in the host cell.

To visualize the viral infection of tissue, Co-Wish simulates the tissue as a lattice in which each node represents one cell. The dimension of the lattice is determined by the user. In our simulations, the host cell division is not designed. In addition, there is no competition between host cells in Co-Wish. Each infected cell can infect its neighboring cells. Co-Wish uses two patterns for determining the susceptible cells; quincunx and square. In the first pattern, an infected cell sits at the center of a quincunx and infects the four neighboring cells in the quincunx (Fig 3—Cross Pattern). In the second pattern, the infected cell is in the middle of a square that also includes the eight susceptible cells around it (Fig 3—Square Pattern). Co-Wish assumes each infected cell is able to infect at most three layers of cells in its vicinity. The proportions of the number of virions in each layer of infected cell vicinity are determined based on a mathematical equation, described in the Methods section.

To select the pattern, we ran different simulations with both quincux and square patterns. The results showed that the two patterns of distributions only affect the pace at which the dynamics of infection unfold and not the observed patterns (Fig 4). Consequently, we used the square pattern that concludes faster dynamics to generate the results.

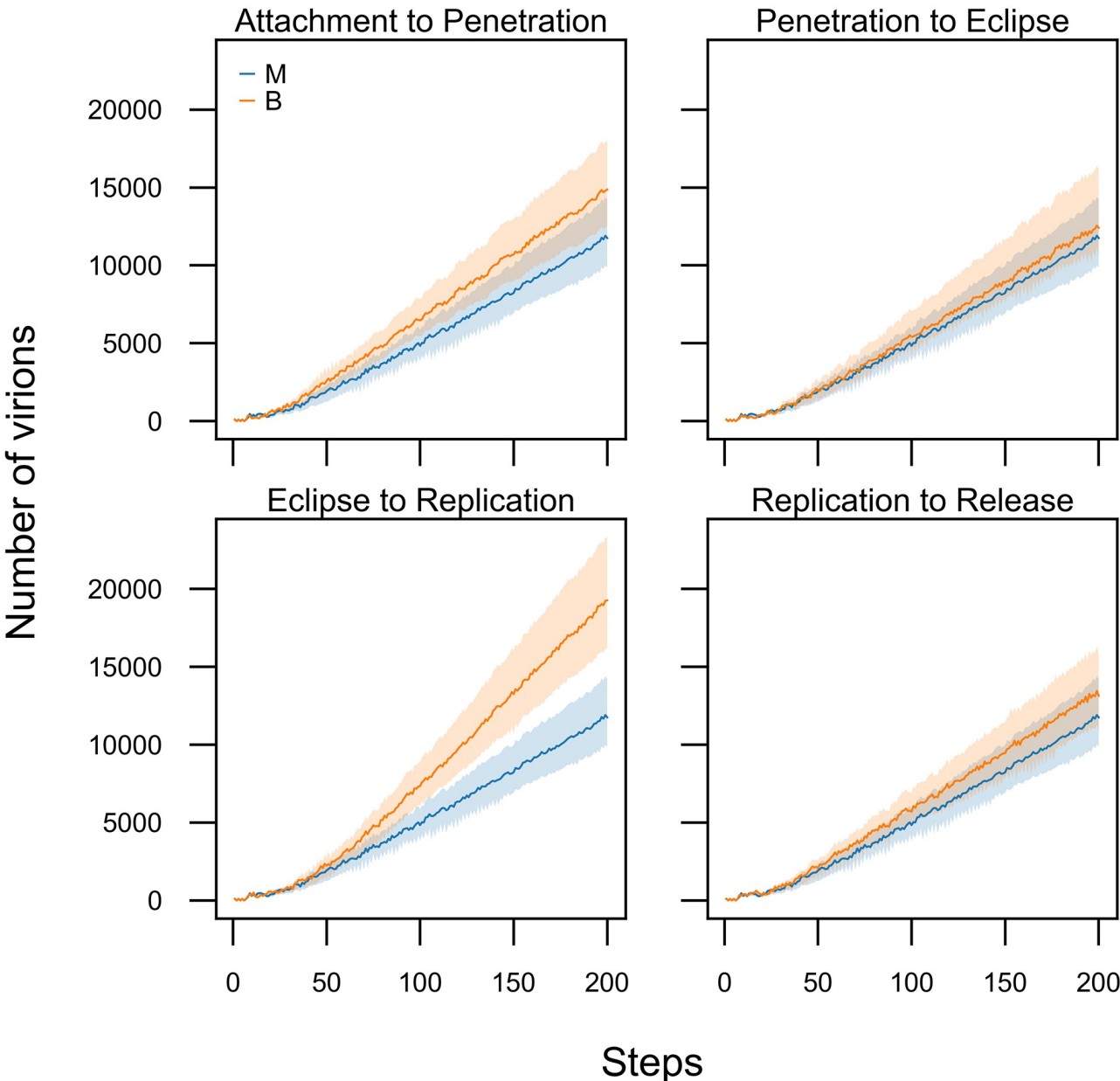

**Fig 2. Changing the waiting times for the different stages of the infection cycle influences the overall dynamic.** In our models, by default generate waiting times from Exponential distribution with one value for its parameter(Exp($\lambda = 1$)), except for the eclipse duration which is generated from Weibull distribution. The scale and shape parameters of the Weibull distribution, both have been set as one(scale = 1, shape = 1). The blue trajectories show the behaviour of our model with the default parameters. To show how changing these parameters affects our model, in each of the four panels, we changed the waiting time of one of the transition (orange trajectories). For the manipulated simulations, the waiting times for transition between attachment, penetration, and replication were generated from Exp($\lambda = 3$). The waiting time of the eclipse stage was generated from Weibull(1, 0.5). Each trajectory is the average of 100 runs. The shaded areas contain the trajectories of all the 100 runs for the given parameters. The immune system was not active in all the runs.

## The influence of the immune system

In Co-Wish, the immune system response is characterized by the probability at which the immune system can eliminate the toxic elements ($\gamma$), the delay between the infection and the immune response, and the capacity of the immune system ($\mathcal{I}$). Toxic elements include one or

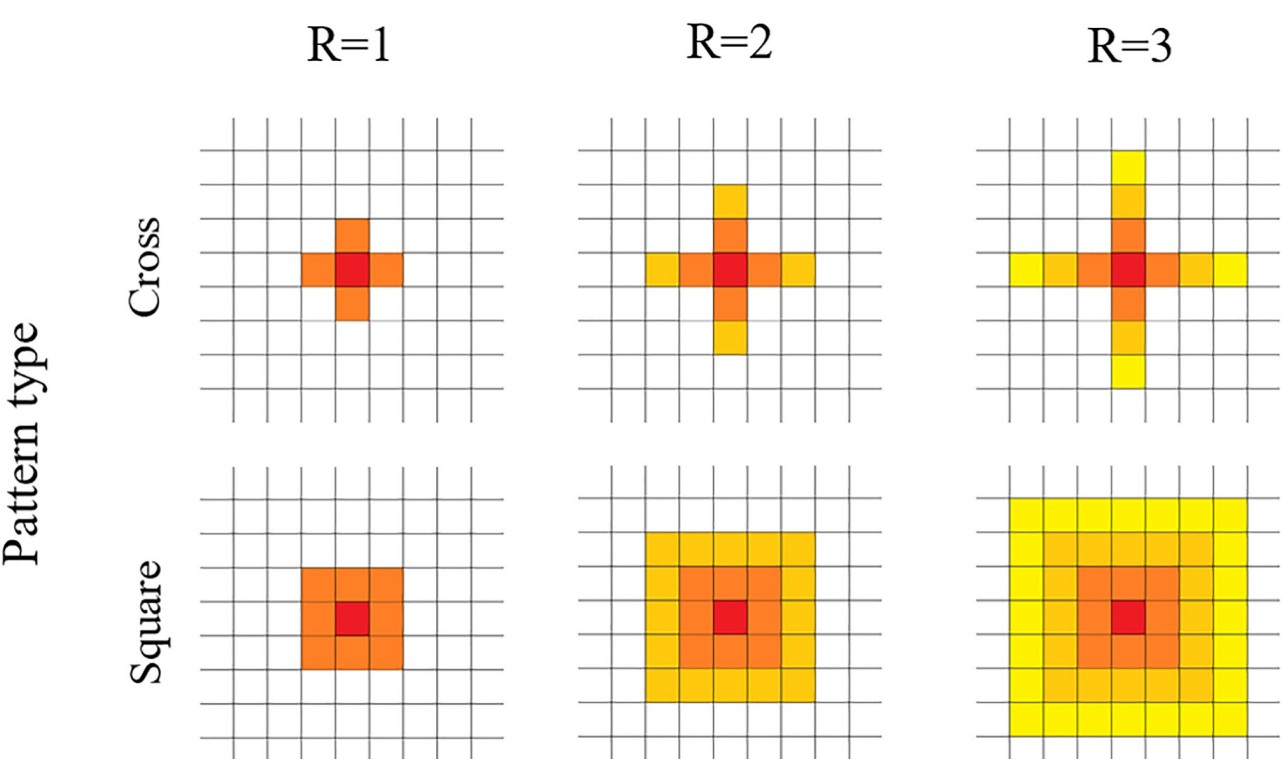

**Fig 3. Viral infection spreading patterns.** In our model, infection can either spread in cross (or quincunx) pattern or a square pattern. The red node demonstrates the infected cell. R determines how many layers of neighbouring cells can be affected by the infected cell in the middle of each pattern.

both of the following: infected cells; virions. In other words, Co-Wish can eliminate the infected cells, the virions, as well as the combination of both. Here, the capacity refers to the maximum number of each kind of toxic element that can be eliminated by the immune system. The capacity value has to be less than the number of toxic elements that are potentially produced in the early steps of the simulation. $\gamma$ is drawn from a uniform distribution, $U(a;b)$, that its parameters are within $[0;1]$. The immune system can eliminate toxic elements with a different probability. This characteristic indicates that all toxic elements produced in each step of the simulation do not have the same probability of elimination. For example, Co-Wish generates separate probability values for the killing of each virion. Here, we assume that a virion is killed with probability $p$ and survives with probability $1 - p$.

After the infection, the immune system mounts a response with a short delay, thus a number of cells become infected before the immune system attempts to combat the infection. The immune system as described here does not distinguish between the innate immune response and the adaptive response, but in general terms, it resembles the innate response since it lacks memory of previous encounters with any of the two strains in our simulations and responds to both pathogens in equal measures.

In our simulations, we employed four broad types of immune response:

1. Weak with limited capacity (WL): $\gamma \sim U(0, 0.4)$

2. Weak with unlimited capacity (WU): $\gamma \sim U(0, 0.4)$

3. Strong with limited capacity (SL): $\gamma \sim U(0.6, 1)$

4. Strong with unlimited capacity (SU): $\gamma \sim U(0.6, 1)$

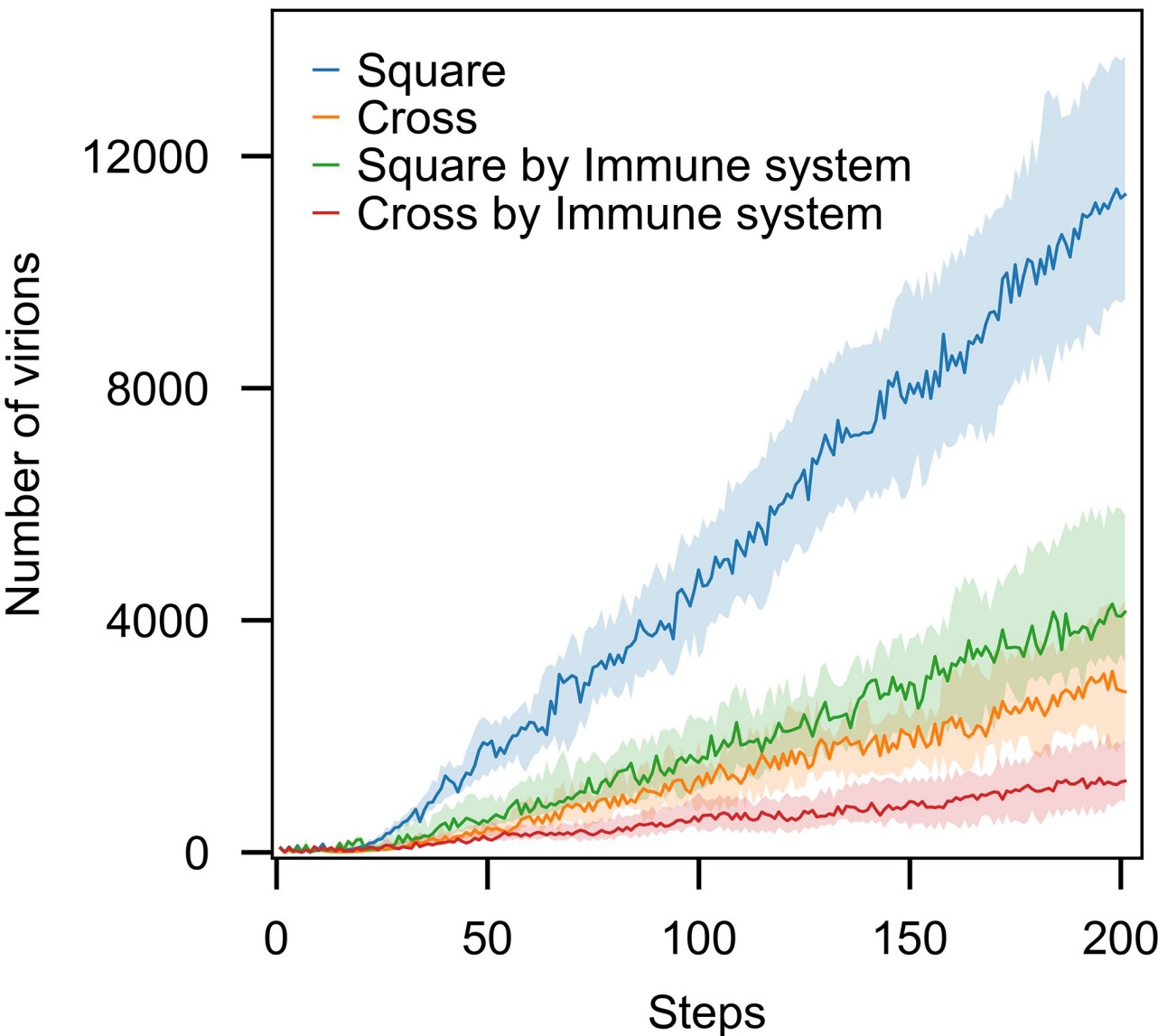

**Fig 4. The patterns of the spread of the infection only affects the pace at which the dynamic of our model unfolds.** Each trajectory is the average of 100 runs. In all the runs, parameters of probability distributions to generate the waiting times were set as one. The shaded areas contain the trajectories of all the 100 runs for the given parameters. The immune system was not active in all the runs.

Limited capacity indicates that the immune system will not be active in all the steps of the simulation. It will be deactivated when the capacity hits the predetermined value for $\mathcal{I}$. If the capacity is unlimited, the immune system will be active by the end of simulation.

Different levels of immune system strength have been studied in a single virus model. Fig 5 represents the number of virions for four levels of the immune system. As expected, decrease the killing rate and putting the restriction to the capacity of the immune system, decrease its strength as well and the slopes of the graphs are steeper for the weaker immune systems.

## Competition during co-infection

In this study, we assumed that the benign strain (virus B) has a shorter infection cycle than the malignant strain (virus M). Therefore, we set the parameters of the model such that the virus B

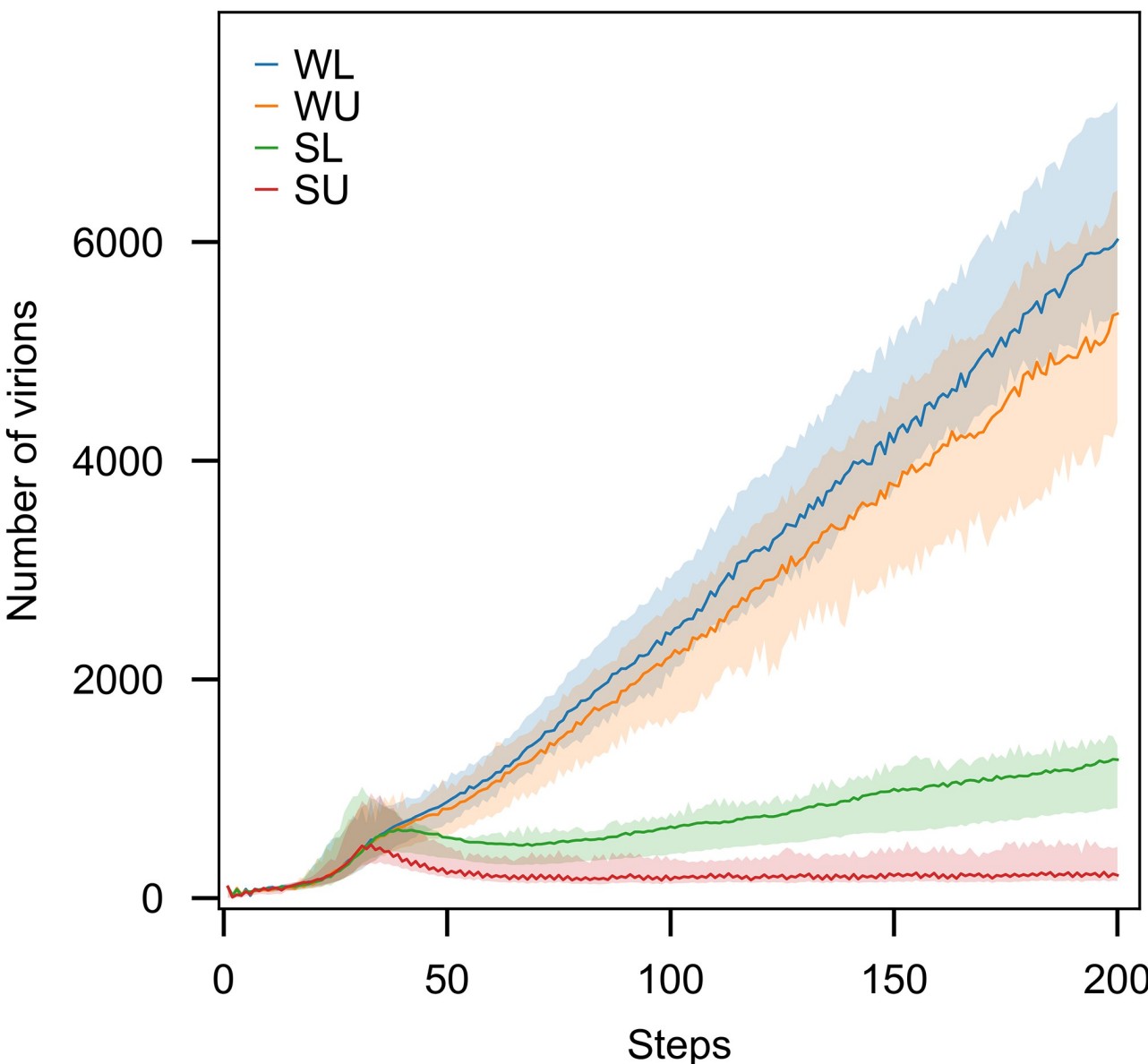

**Fig 5. The number of virions produced during the infection is affected by the strength of the immune system.** We utilised four different modes of immunity in our model: weak and limited capacity (WL), weak and unlimited capacity (WU), strong and limited capacity (SL), strong and unlimited capacity (SU). Each trajectory is the average of 100 runs. In all the runs, parameters of probability distributions to generate the waiting times were set as one. The shaded areas contain the trajectories of all the 100 runs for the given parameters.

spends less time in each of the four stages of viral growth—from attachment to replication—compared with the virus M. In addition, we assumed that the burst size of virus B is less than virus M, due to its shorter waiting time in the replication stage. The location of the second virus in the lattice is chosen randomly, and it is independent of the first virus location. Those cells that are not infected by the first virus can be a niche of the second virus.

To simulate the co-infection, we considered three scenarios:

1. Two viruses infect the host simultaneously.

2. The host is infected by B and later by M

**Table 1.** (*): $c = 100$; The parameter $p$ has been calculated according to Eqs (1) and (2); to calculate $P1$, the time $T$ has been set as 5. (**):$P1$ has been calculated according to Eqs (4) and (5). The parameters $\beta_0$ and $\beta_1$ in Eq (4) have been set as zero and 1, respectively.

| virus | Viral Growth Stage | Waiting Time Distribution | Transition Probability Distribution |
|---|---|---|---|
| B | Attachment-Penetration | $exp(\lambda); \lambda > 1$ | Beta(1,1) |
| | Penetration-Eclipse | $exp(\lambda); \lambda > 1$ | $Beta(1, 1)$ |
| | Eclipse-Replication | $weibull(\lambda = 1, \omega); \omega < 1$ | $Beta(cp, c(1 - p))^*$ |
| | Replication-Shedding | $exp(\lambda); \lambda > 1$ | $P1^{**}$ |
| M | Attachment-Penetration | $exp(\lambda = 1)$ | $Beta(1, 1)$ |
| | Penetration-Eclipse | $exp(\lambda = 1)$ | $Beta(1, 1)$ |
| | Eclipse-Replication | $weibull(\lambda = 1, \omega = 1)$ | $Beta(1, 1)$ |
| | Replication-Shedding | $exp(\lambda = 1)$ | $P1^{**}$ |

3. The reverse of second scenario

Co-infection delay $\mathcal{D}$ refers to the time delay between the introduction of each viral strain to the tissue. In the first scenario, the co-infection delay is zero. If the co-infection delay is set as 30 steps, it means the second virus enters into the model thirty steps after the first virus. During this time, the first virus starts to infect some cells. In Co-Wish, only one strain enters a cell—thus, cells are not co-infected—but the pattern of the spread of each strain in a co-infected fashion affects the spread of the strain by limiting the susceptible cells in the tissue.

To test the three scenarios of co-infections in our model—namely, (i) no delay between the introduction of M and B strains, (ii) B is introduced first, and (iii) M is introduced first—we simulated each scenario (using parameters in Table 1). The co-infection delay ($\mathcal{D}$) in each scenario, was set to 0, + 30, and −30 steps, respectively. In the absence of the immune response, the benign strain is always dominant (Fig 6). The results suggest that the waiting times greatly influence the dynamics of co-infection. While the B strain spends less time in the replication stage and, consequently, produces a smaller burst compared to the M stain, its shorter infection cycle enables it to out-compete the M stain during co-infection.

Since the main idea of this work is to explore the theoretical feasibility of viral co-infection of two malignant and benign strains to reduce the spread of malignant one, we simulated the third scenario—B is introduced first—separately by changing the parameters of applied distributions to generate the waiting times. We considered four conditions. In each condition, the waiting times have been shortened by increasing the value of Exponential distribution parameters and decreasing the shape parameters of Weibull distribution. In other words, we investigated how changing the waiting times can influence the number of steps at which the B strain out-competes the M strain (Fig 7). The results show that reducing the waiting times decreases the time it takes for the B strain to out-compete the M strain (Fig 8).

## Discussion

In this study, we look into the dynamics of viral co-infection using a CEP-inspired computational model. To explore the possible effects of co-infection by two virus species, we develop a stochastic-based viral infection simulator—called Co-Wish—to model and display the competition between two strains. Co-Wish offers a flexible framework to study the stochastic effects induced during the evolution of the infection. Co-Wish supports simulating both discrete and stochastic behaviors of each stage of viral growth. It explicitly simulates the behavior of each virus in each stage of its growth process. Co-Wish also incorporates the effect of the innate immune system on the dynamics of viral infection.

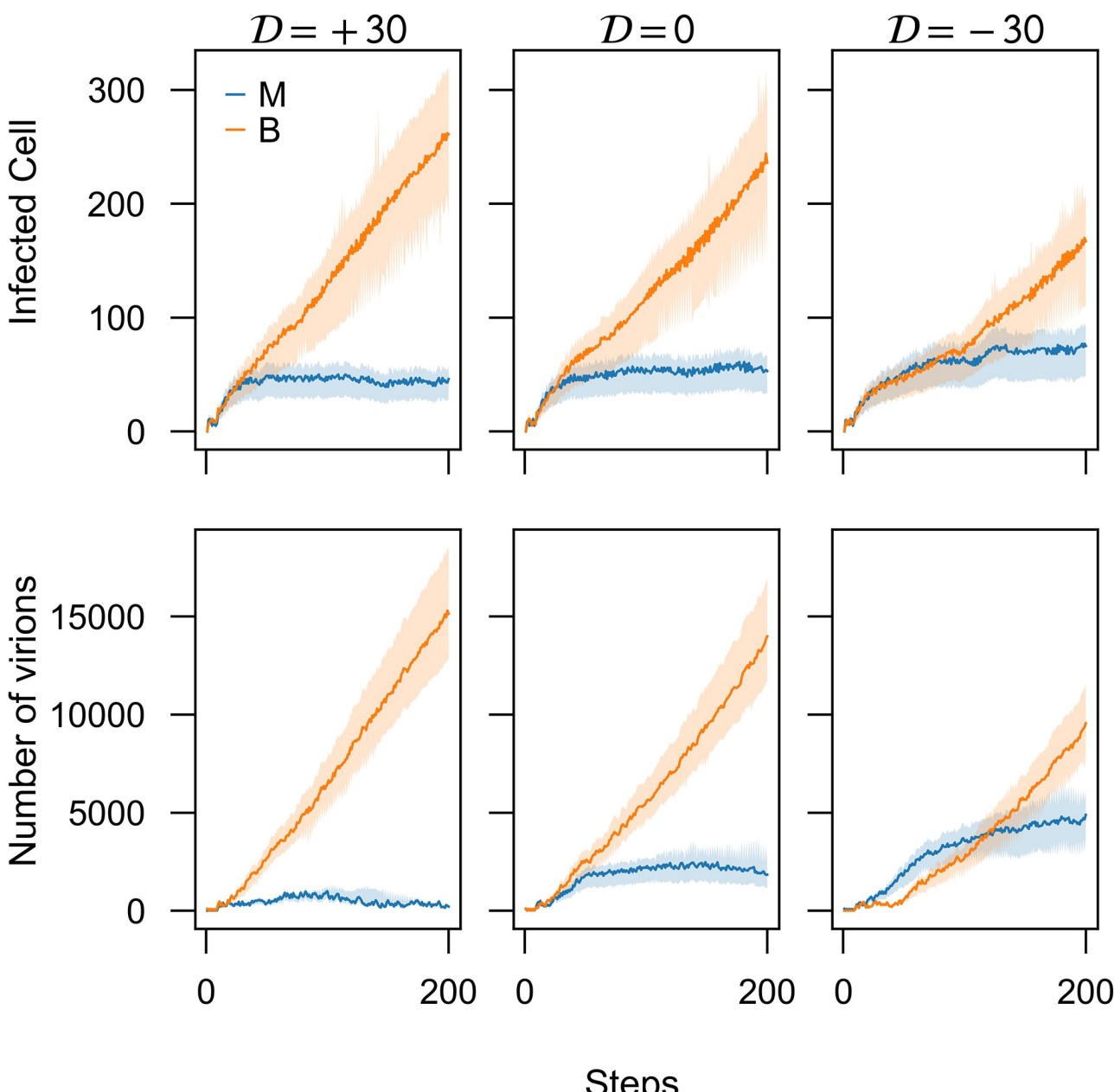

**Fig 6. Benign strain is dominant in all three conditions of co-infection delay** ($\mathcal{D}$)**.** Introduction of the benign strain (B) 30 steps before the introduction of the malignant strain ($\mathcal{D} = +30$), or simultaneously ($\mathcal{D} = 0$) greatly hampers the spread of B strain but introducing this strain 30 steps after the introduction of M strain ($\mathcal{D} = -30$) tighten the competition between the strains. Each trajectory is the average of 100 runs. The shaded areas contain the trajectories of all the 100 runs for the given parameters. For virus M, all the parameters of probability distributions to generate thewaiting times were set as one. For virus B, the waiting times had been generated from $\text{Exp}(\lambda = 3)$and $\text{Weibull}(\lambda = 1, \omega = 0.5)$. The immune system was not active in all the runs.

Several studies consider the concept of CEP to illustrate the competition between viruses [21–23]. In particular, Peter and colleagues in [24] recently presented a survey on SARS-CoV-2 infection dynamics models, focusing mostly on ODE-based methods. Typical models of virus dynamics are developed based on three ordinary differential equations concentrating on the number of target cells, free virions, and infected cells [15]; i.e., most of these approaches

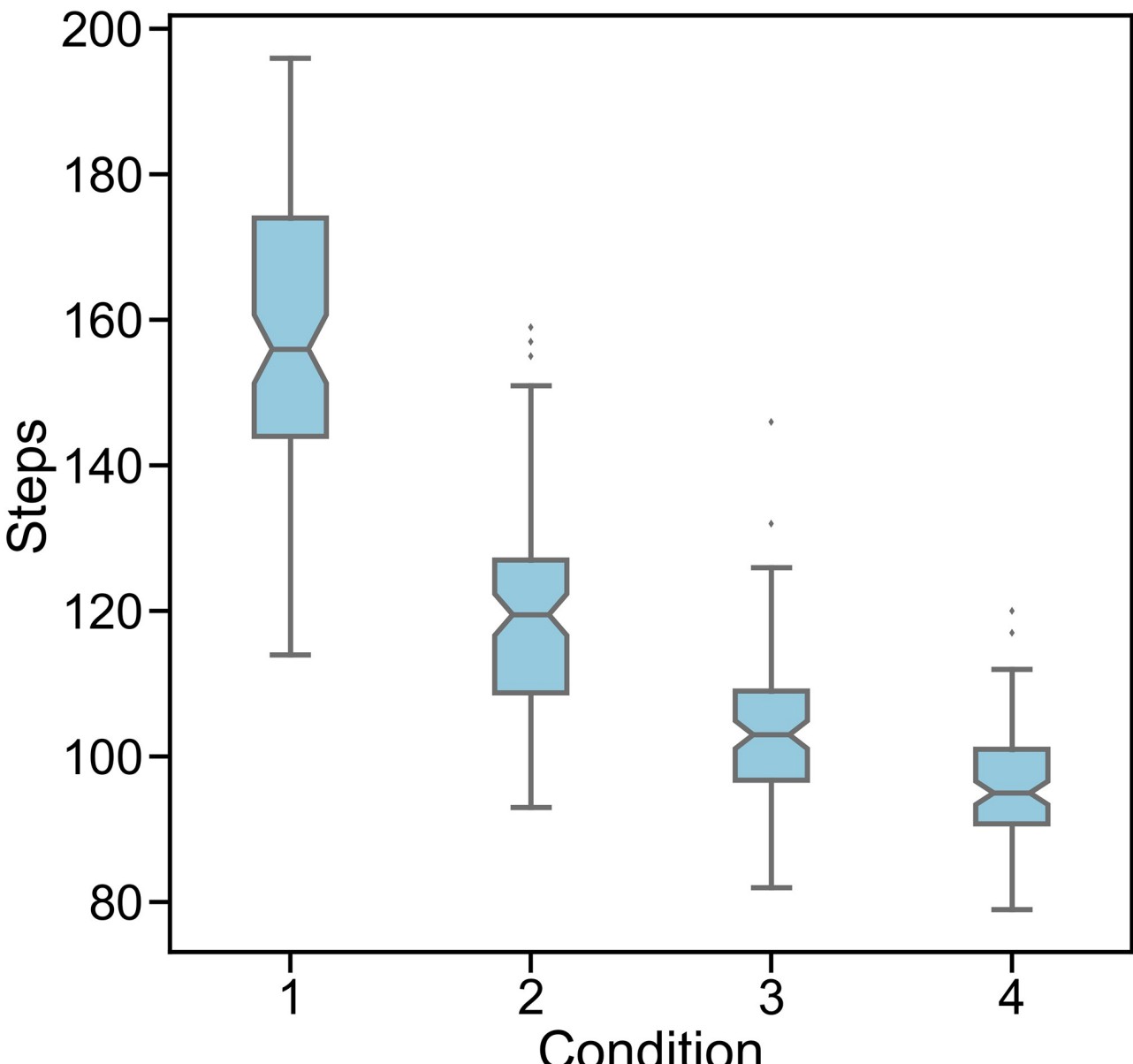

**Fig 7. Changing the parameters of probability distributions to generate waiting times affects the number of overtaking step of benign strain when co-infection delay is (-30).** The parameters include $\lambda$, the parameter of Exponential distribution and $\omega$, the parameter of Weibull distribution. Condition 1: $\lambda = 2$, $\omega = 1/2$; condition 2: $\lambda = 3$, $\omega = 1/3$; condition 3: $\lambda = 4$, $\omega = 1/4$; condition 4: $\lambda = 5$, $\omega = 1/5$. Boxes show the median and first and third quartiles and the notches represent the %95 confidence interval around the median. The immune system was not active in all the runs.

proposed for mathematical modeling of host-pathogen interactions can be seen as an extension of the ODE-based model [25–27]. Although these extended models provide explainable frameworks in which they improve our understanding of the dynamic of the viral infection within the host, they somehow suffer the main issue about deterministic models, namely, relying on viruses' average behaviors during the infection process [16, 17]. As a result, ODE-based approaches often ignore the stochastic nature of the viral infection. Therefore, controlling the heterogeneity of virus populations affecting competitive exclusion is not readily achievable in the ODE-based methods. Accounting for the viral infections stochastic nature during the

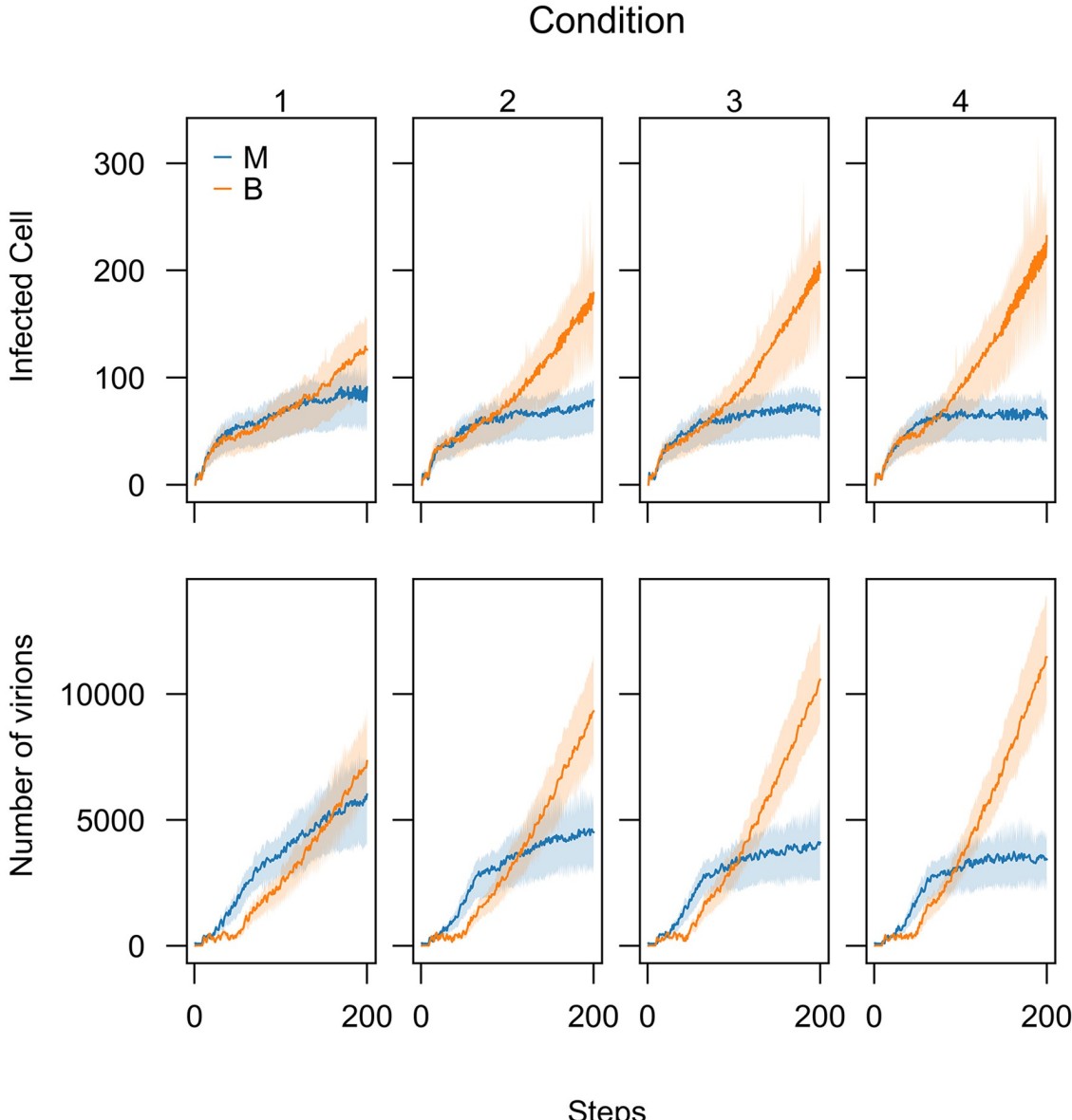

**Fig 8. Different waiting times expedites benign strain (B) to outcompete malignant strain (M) when co-infection delay is (-30).**
To generate different waiting times for virus B growth process, $\lambda$–the parameter of exponential distributions–and $\omega$–the shape parameter of the Weibull distributions–have been changed. Condition 1: $\lambda = 2$, $\omega = 1/2$; condition 2: $\lambda = 3$, $\omega = 1/3$; condition 3: $\lambda = 4$, $\omega = 1/4$; condition 4: $\lambda = 5$, $\omega = 1/5$. For virus M, all the parameters of probability distributions to generate the waiting times were set as one. The immune system was not active in all the runs.

simulation process, in contrast, is one of Co-Wish's most essential features, enabling our model to demonstrate the competition between virus strains more realistically.

Using Co-Wish, we first simulate the characteristics of two virus growth processes separately. Then, we investigate their interactions until one can dominate the other. To model the co-infection of two viruses using Co-Wish, we consider three scenarios including (i) two viruses infecting the host at the same time,(ii) delayed initiation of the malignant-virus infection, and (iii) delayed initiation of the benign-virus infection. While an earlier stage of infection initiation for one of the viruses can be considered a competitive advantage, it plays a

minor role as compared to the waiting times in the five stages of the viral infection of a cell in the proposed model—namely attachment, penetration, eclipse, replication, and release. In summary, we observed that these waiting times in the five stages of the viral infection—regardless of varying the starting times for the viruses—crucially affect the competition between two viruses over a shared host. In other words, if the starting time for the benign virus with a higher viral growth rate is not delayed too long, it can eventually block the infection caused by the malignant virus. In particular, our results showed that the waiting times at the eclipse stage significantly affected the number of virions in the competition between the two viruses. We also observed a similar trend when we incorporated an innate immune response during viral co-infection.

The dynamics of viral co-infection remain one of the lesser-explored areas of virology and epidemiology. The model presented in this paper illustrates how the competition between viral strains over a shared niche can hinder the propagation of one, an unsurprising result given the competitive exclusion principle. While the model is inescapably a simplified reconstruction of what might happen in nature, a few lines of experimental studies discussed below provide some credence to the hope that the dynamic of co-infection can be employed to control the spread of lethal strain by introducing a more benign strain, but many obstacles must be overcome before this hope can become a viable epidemiological strategy.

Previous studies have investigated the association between co-infection with the GB virus C and survival among individuals with HIV infection [28, 29]. For example, George and colleagues reported a significantly higher survival rate among HIV-positive patients co-infected with the GB virus C virus than among HIV-positive individuals without GB virus C co-infection [28]. This result—a beneficial effect of co-infection with the GB virus C virus on HIV-related survival—suggested the GB virus C infection as a potential treatment for HIV-infected patients [28]. Our CEP-based model also suggests that one viral strain might be able to outcompete and eliminate a rival strain in the same ecological niche during a co-infection scenario. Suggestively, our theoretical framework hints at an intriguing research direction in the dynamics of co-infection, viz. the possibility of controlling the spread of any viral infection in the population by supplanting the lethal virus with a much less fatal one. For example, researchers could employ our theoretical model to explore and evaluate the CEP-inspired hypothesis in tackling a viral respiratory pandemic, such as COVID-19, through co-infection with some forms of common respiratory viruses that are less harmful. However, in the case of COVID-19, it could prove challenging to find such a harmless virus that would keep SARS-Cov-2 below the exposure level. Also, there is no guarantee that—even if our proposed CEP-based hypothesis works—SARS-Cov-2 does not come up as the winner of such competition. Nevertheless, there are four harmless human Coronaviruses—namely HCoV-NL63, HCoV-229E, HCoV-HKU1, and HCoV-OC43—that lead to a range of respiratory diseases with mild symptoms—cause approximately 15% of common colds [30]. Recent studies [31, 32] also have revealed that hospitalized COVID-19 patients who developed severe symptoms of COVID-19, such as high fever and pneumonia, were rarely co-infected with other common respiratory viruses, particularly the common colds. From the immune system's point of view, this might be related to the result of a new study [33] that discusses that COVID-19 patients previously infected with other forms of the human beta-coronaviruses could also develop some types of immunity to SARS-CoV-2. Yet, further experimental investigations are needed to understand the therapeutic potentials of viral co-infection thoroughly. In particular, there is an urgent need to establish clinical trials assessing the role of simultaneous respiratory viral infection in the variability of disease virulence among COVID-19 patients.

It is also worth mentioning that the notion that the dynamics of viral co-infection in respiratory diseases can be dominated by competition was previously highlighted by Burattini and

colleagues [21] and Pinky and Dobrovolny [19]. The former resorts to CEP as a premise to investigate the conditions under which two or more strains can coexist within a host, while in the latter, Pinky and Dobrovolny explored the possible competition between influenza A virus and a handful of respiratory viruses [19]. Although their premise is the aforementioned studies start from a postulation similar to our work, their approach—in modeling the co-infection dynamics using ODEs—is quite different from the one applied in this work. In particular, we modeled the infection at the cell level. Also, our model—as opposed to the one proposed by Burattini and colleagues and Pinky and Dobrovolny—includes the immune response Neither of these main differences is expected to profoundly change the qualitative behavior of two viral strains competing over an ecological niche. Nevertheless, we believe that the added complexity of the model is a step in the right direction. Thus, the present study can be considered the logical extension of previous works on this topic.

There are several important limitations to the idea behind our approach. Crucially, it would be deeply preposterous to suggest a model of viral co-infection as a basis for prescribing remedies for public health issues, such as the COVID-19 pandemic. Anyone with a cursory knowledge of biology, specifically epidemiology and ecology, understands the limitations of even the best and most detailed models in predicting the interaction between living entities. Rather, introducing the notion of Gause's law—a quintessentially ecological concept—to the dynamics of co-infection is one of the first steps to explore the therapeutic ramifications of the CEP as applied to viral co-infection. The journey from this simple theoretical curiosity to a usable remedy against viral epidemics undoubtedly will have to be paved by a thick layer of theoretical and experimental studies. Given that many characteristics of viral life cycles remain elusive (e.g., the burst size of different viral strains), applying Gause's law to combat viral epidemics will likely remain a fanciful notion for many years to come.

## Methods

Co-Wish—the proposed model—is a stochastic-based model to simulate the stages of viral infection. To model the viral infection, we consider each stage of infection as a probabilistic process. Each stage is simulated by applying Gillespie's stochastic algorithm [20].

### Viral growth process

To simulate the viral growth process, Co-Wish considers the duration of each state (waiting time) and the probability of transition from one state to the next one (transition probability) The waiting times follow either an exponential, $Exp(\lambda)$, or a Weibull distribution, $W(\lambda, \omega)$. Weibull distribution is a widely used distribution to generate time-to-failure data. If variable $t$ is defined as time-to-failure, the Weibull distribution describes the hazard rates as being proportional to a power of time. It is worth pointing out that in survival analysis, the event of interest is typically referred to as a 'Failure.' The hazard rate is the potentiality that an event will occur within an immeasurably short time interval (between time $t$ and $t + \Delta t$, as $\Delta t$ goes to 0), given that the event has not yet occurred—i.e., the instantaneous risk that an event will happen to a particular patient at a certain time, given that the patient at risk has survived up to that time. In the Weibull distribution, the hazard function is $h(t) = \lambda \omega t^{\omega-1}$, where $\lambda > 0$ and $\omega > 0$. The parameter $\omega$ —called a *shape parameter*— determines the shape of the hazard function. If $\omega > 1$, then the hazard increases as time increases. If $\omega = 1$, then the Weibull distribution collapses to an Exponential distribution in which the hazard function is constant ($h(t) = \lambda$). If $\omega < 1$, then the hazard rate decreases over time [34]. Applying the Weibull distribution allows us to control the hazard rate of transition to the next stage by determining different values for the shape parameter $\omega$.

All transition probabilities are generated from the Beta distribution. The parameter values of the Beta distribution depend on the chosen distribution for the waiting time of the current stage. If the waiting times of one stage are generated from an Exponential distribution, the corresponding transition probabilities will be generated from $Beta(1, 1)$. If waiting times are drawn from a Weibull distribution—inspired by the survival analysis concepts—the transition probabilities will be generated based on the probability of remaining in the current stage [34]).

The survival function for the Weibull distribution determines probability of remaining in a given stage [34], i.e., the transition into the next stage is our event of interest and the time **T** is defined as the maximum time it takes for a virus to transmit into the next stage. Then, we use the survival function ($S(T)$) for Weibull distribution to determine the probability of transition from the current stage to the next one $P$:

$$S(T) = exp(-\lambda T^{\omega}), \quad P = 1 - S(T) \quad . \tag{1}$$

We consider $P$ as the expected value of the Beta distribution ($E(t)$) and determine the parameters of Beta distribution according to the value of $P$:

$$E(t) = \frac{\alpha}{\alpha + \beta} = P \tag{2}$$

To estimate the parameters, we keep the sum of the parameters constant, i.e., $\alpha + \beta = C$. Consequently, the parameters of the beta distribution will be

$$\alpha = CP, \quad \beta = C(1 - P) \quad . \tag{3}$$

In addition, to avoid getting zero values for the parameters, when $P$ is either zero or one, a positive small value ($\epsilon$) is added to the above estimation.

To determine the probability of transition from the replication stage to the release stage, we assume that the probability is a function of the number of viruses produced during the replication stage. In other words, We consider the transition of replication stage to the release stage as a binary event. We assume a linear relationship between the number of produced viruses in the cell($x$) and the log-odds of transition from the replication stage to the shedding stage:

$$\log_b \frac{p_{rs}}{1 - p_{rs}} = \beta_0 + \beta_1 x \quad , \tag{4}$$

where $p_{rs}$ is transition probability from replication stage into the shedding stage. Therefore,

$$p_{rs} = \frac{b^{\beta_0 + \beta_1 x}}{1 + b^{\beta_0 + \beta_1 x}} \quad . \tag{5}$$

We suppose increasing the number of produced virions in the replication stage, increases the log-odds of transitioning to the release stage. Therefore, the positive value should be determined for parameter $\beta_1$. A positive value for $\beta_1$ means that increasing $x$ by one increase the log-odds by $\beta_1$. The number of viruses produced in a cell depends on the waiting time of the replication stage, i.e., longer waiting times results in the production of more viruses. During the replication stage, the virions are replicated according to stochastic lags. These lags between replication of each virus is determined stochastically. The replication process continues until these lags add up to the total waiting time of the replication stage.

### The spread of viral infection

After determining the patterns of infection, we approach to define the proportion of number of virions attached to each susceptible cell in three layers in its vicinity ($V_i; i = 1, 2, 3$):

$$V_i = \frac{k_i V}{n_i} \quad , \tag{6}$$

where $V$ is the number of virions released from an infected cell and $n_i$ is the number of susceptible cells in each layer of cells surrounding the infected cell, according to either cross or square pattern. In the square pattern, $n_1$, $n_2$, and $n_3$ are 8,16, and 24, respectively. $k_i$ is a weight designating what shares of produced virions belong to each layer of vicinity. For example, if the total number of released virions from one cell is 1000 and $k_1 = 0.4$, $k_2 = 0.3$, and $k_3 = 0.3$, 400, 300, and 300 virions will be belonged to each layer, respectively. In each layer, the virions are diffused uniformly among cells.

### Infection calculation: Number of virions and infected cells

The number of infected cells and free virions is calculated in each step of the simulation. The time length of each step is predefined before starting the simulation. Hence, the time length of the steps is constant. In our study, the time length of the simulation steps is about 1000 milliseconds. Co-Wish runs a number of Gillespie algorithms simultaneously. In other words, many cells can be infected at the same time.The time lengths of the algorithms with different settings are not alike. Each step is terminated when the sum of the time length of different runs of Gillespie algorithm is equal to the determined time length of the step.

## Supporting information

**S1 File.**
(PDF)

## Author Contributions

**Conceptualization:** Maryam Shahdoust, Mehdi Sadeghi.

**Methodology:** Safar Vafadar, Maryam Shahdoust.

**Project administration:** Mehdi Sadeghi.

**Resources:** Pooya Zakeri.

**Software:** Safar Vafadar.

**Supervision:** Mehdi Sadeghi.

**Validation:** Mehdi Sadeghi.

**Visualization:** Safar Vafadar, Ata Kalirad.

**Writing – original draft:** Maryam Shahdoust, Ata Kalirad, Pooya Zakeri.

**Writing – review & editing:** Maryam Shahdoust, Ata Kalirad, Pooya Zakeri.

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
