## [Decision Letter · Decision Letter 0]

23 Sep 2020

PONE-D-20-22300

Competitive exclusion during co-infection as a strategy to prevent the spread of a virus: a computational perspective

PLOS ONE

Dear Dr. Sadeghi,

Thank you for submitting your manuscript to PLOS ONE. After careful consideration, we feel that it has merit but does not fully meet PLOS ONE’s publication criteria as it currently stands. Therefore, we invite you to submit a revised version of the manuscript that addresses the points raised during the review process.

We look forward to receiving your revised manuscript.

Kind regards,

Bashar Ibrahim

Academic Editor

PLOS ONE

Journal Requirements:

Reviewers' comments:

Reviewer's Responses to Questions

**Comments to the Author**

1. Is the manuscript technically sound, and do the data support the conclusions?

Reviewer #1: Yes

Reviewer #2: Partly

Reviewer #3: Partly

2. Has the statistical analysis been performed appropriately and rigorously? 

Reviewer #1: Yes

Reviewer #2: N/A

Reviewer #3: N/A

3. Have the authors made all data underlying the findings in their manuscript fully available?

Reviewer #1: Yes

Reviewer #2: Yes

Reviewer #3: Yes

4. Is the manuscript presented in an intelligible fashion and written in standard English?

Reviewer #1: Yes

Reviewer #2: Yes

Reviewer #3: No

5. Review Comments to the Author

Reviewer #1: Dear authors,

please notice the following comments and suggestions regarding your paper and correct the regarding parts if necessary:

(1) lines 101 - 103: Is there a word missing?

(2) caption of Fig. 2: "The each trajectory..." This sentence does not make sense. It appears several times in the paper !!!

(3) !!! Almost all references to the Figure numbers in the text seem to be wrong, i.e. shifted by one !!!

(4) line 178: "strong With limited capacity" - "w" is missing

(5) lines 224 - 226: The plot does not really look LINEAR. How do you draw this conclusion. It could also be a quadratic or exponential decline.

Reviewer #2: This study tries to assess the feasibility of use of co-infecting virus to combat problematic virus, based on simulation results. The main conclusion obtained by the authors is that if co-infecting (harmless) virus can accumulate more rapidly than the target (harmful) virus and if the two viruses compete for the same type of host cells, the target virus will be dominated by the co-infecting virus, even when the co-infecting virus came later; to be honest, this conclusion will not surprise anyone, but I can find a potential value of this study in establishment of a framework for discussion on “under what circumstances such antiviral strategy can function.” To fulfill the potential value of the study, I recommend the authors to consider the following points:

Major points:

1) The simulation results are totally dependent on parameter values, especially on lambda. Please provide lambda (and other parameter) values used for the simulations explicitly in every figure; to avoid confusion, clearly define parameters. For example, define different lambda value for each virus by using lambda B and lambda M.

2) Please provide average waiting times for each step, as readers can compare them with D.

3) Figures and figure legends do not correspond to each other in some instances. I need explanation for Fig. 3 but I wonder where I can find it...

4) In Fig. 6, B virus continues to accumulate but M virus does not. Could the authors provide intuitive explanation of this phenomenon? For example, limitation in the number of host cells will not explain because B virus would also suffer.

5) In relation to the above comment, I’m not sure if the authors assumed (1) limitation in the number of cells, (2) host cell division, and (3) innate immunity in each figure.

6) In my understanding, authors think that competition for host cells is important in the simulation. Could the authors provide the time-course change of the number (and/or proportion) of cells attacked by both of the virus (before they get infected)? I’m not sure if the local spread of viruses assumed in this manuscript allows competition for host cells, though it totally depends on the virus and host cell densities assumed.

7) L40 and L312: the authors touch incubation times of viruses but it was not clear to me how they are relevant to this study.

Other minor points:

8) Figure 2 Legend: Replication -> replication

9) L178: ith -> with

10) The reference formatting is terrible.

Reviewer #3: In this paper the authors have presented an interesting framework for the progression of viral infection in host tissue. The model simulates the invasion of two competing viruses in a theoretical lattice of initially susceptible cells. Cells that are infected by one virus are prevented by being simultaneously infected by a competing strain to simulate the effect of the competitive exclusion principle (CEP). The premise of this paper therefore may be appealing to not only mathematical modellers but also biologists and medics working in related fields.

However, I do have some key concerns, that if addressed would benefit the manuscript:

- The authors do not provide a strong justification for the application of their model to SARS-CoV-2 in their introduction, which detracts from the narrative since the reader is left questioning the relevance of the model in this setting. Some justification is provided later in the discussion section, however, this is not particularly strong, and thus the application of the model to SARS-CoV-2 feels very extrapolative. I would therefore encourage the authors to consider removing the emphasis of the model’s application to SARS-CoV-2 in the abstract and introduction of the paper.

- The action of the CEP in the model was buried in the paper (it was briefly alluded to in lines 201 -203). This could have been made much clearer. Moreover, it appears that the authors are more specifically modelling superinfection exclusion, whereby secondary infection is prevented at a cellular level, which I would consider to be a specific example of the CEP.

- The paper would benefit if the model was described in clearer detail. For example, the authors have not made the action of the immune response in the model clear. I appreciate that to appeal to a wider audience that the authors may not wish to go into a great level of detail in the main body of the text, but I feel a more detailed description of all of the components of the model in the supplementary information would be very beneficial.

- The paper would benefit from a more thorough justification of the parameterisation of the model, with suitable references to existing literature. While I appreciate different parameter combinations have been explored, this is somewhat limited, and I feel a more detailed sensitivity analysis would be beneficial. This also would most likely be appropriate to include in the supplementary information.

- The authors do not appear to use time units in their parametrisation of the model, or in the results presented, where all figures are presented in “steps” along the x-axis.

- The above point raises significant concerns – since the model is based on a Gillespie algorithm, each time step will be of a different time length due to different sojourn times between events. It could therefore be misleading to plot the trajectories of their results (e.g. number of virions) against the step count for each run of the simulation, since these are evolving over different timescales. I would therefore strongly advice that the model is revisited to ensure that all x-axes are presented over some units of time (hours or days).

- Additionally, a thorough spelling and grammar check is required

While I do have a significant number of concerns with the manuscript in its current form, I do believe that a valuable piece of work could result from the model framework that the authors have used. Moreover, I felt the premise of the model is interesting in principle and would like to thank the authors for sharing their work.

I have made some additional inline comments below:

line 69 - 70: If there is evidence of a reduced risk of a subsequent coronavirus infection by another respiratory virus then I think this needs to be explicitly stated to be able to back up the paper's justification for applying the model to SARS-CoV-2. Currently, it's not specified which pairs of viruses had a negative interaction between them in ref [10].

Line 133: The legend for figure 2 does not appear to match figure 2 in this document, which appears to show the different spreading patterns (square vs cross)

line 165: Does "the capacity of the immune system (I)" have any units, such as the maximum number of infected cells/virions eliminated per day? It is not clear how "I" is defined mathematically in the model.

line 167: Gamma appears to have been defined in lines 163-164 as "the rate at which the immune (system) eliminates infected cells or virions". Therefore, it presumably has units of "cells/virions cleared per hour/day". I therefore do not understand the logic behind why it is then drawn from a U[0,1] distribution.

Line 176: to differentiate this from the "unlimited immune response capacity" situation, I think it would be better to have notation along the lines of 0>i<i_c bound.="" i_c="" is="" some="" upper="" where="">

line 199: It is unclear how the second virus is seeded into the model. It would be helpful if the authors clarified this (e.g. is it seeded into a cell in the lattice at random?)

line 213: There is a lack of justification for the parameters chosen for the various distributions in table 1. While there may be limited literature for SARS-CoV-2, literature on other respiratory viruses may allow the authors to justify their choices. As highlighted before, I feel the paper would benefit if all times are given in appropriate units such as days, rather than being treated as dimensionless. I would also encourage the authors to consider including a sensitivity analysis of the different parameters in the supplementary information.

lines 267-271: Hepatitis G is now typically referred to as GB virus C

lines 312-314: If the authors wish to keep the focus of the paper on SARS-CoV-2, I feel the manuscript would benefit if these studies (refs [30] and [31]) were highlighted in the introduction section. The reader would then understand the authors justification for the model to be placed in the context of SARS-CoV-2.>/i<i_c>

6. PLOS authors have the option to publish the peer review history of their article (what does this mean?). If published, this will include your full peer review and any attached files.

Reviewer #1: No

Reviewer #2: No

Reviewer #3: **Yes: **Andrew C Glover

---

## [Author Response · Author response to Decision Letter 0]

30 Oct 2020

Point-by-Point responses to Reviewer #1

Dear authors,

Please notice the following comments and suggestions regarding your paper and correct the regarding parts if necessary:

We would like to thank the reviewer for the valuable comments and suggestions. We revised and improved our manuscript accordingly. The responses to the comments are as follows.

C1-1) lines 101 - 103: Is there a word missing?

A1-1) It is now edited.

C2-1) Caption of Fig. 2: "The each trajectory..." This sentence does not make sense. It appears several times in the paper!!!

A2-1) They are all edited now.

C3-1) Almost all references to the Figure numbers in the text seem to be wrong, i.e. shifted by one!!!

A3-1) We are sorry for the confusion we have caused. The figure numbers are now corrected.

C4-1) Line 178: "strong With limited capacity" - "w" is missing.

A4-1) It is now edited.

C5-1) Lines 224 - 226: The plot does not really look LINEAR. How do you draw this conclusion? It could also be a quadratic or exponential decline.

A5-1) The main idea of the plot is to display the effect of shortening the viral growth process, by generating shorter waiting times, on the time at which the benign virus out-competes the malignant virus. We did not study the statistical association which is possible between the value of the probability distribution parameters and the out-competing time. In the revised version of the manuscript, we have omitted the sentence which had made misunderstanding about the kind of correlation between probability distributions parameters and the benign virus overtaking time. The text in lines[225]-[234] now reads:

“Since the main idea of this work is to explore the theoretical feasibility of viral co-infection of two malignant and benign strains to reduce the spread of malignant one, we simulated the third scenario—B is introduced first—separately by changing the parameters of applied distributions to generate the waiting times. We considered four conditions. In each condition, the waiting times have been shortened by increasing the value of Exponential distribution parameters and decreasing the shape parameters of Weibull distribution. In other words, we investigated how changing the waiting times can influence the number of steps at which the B strain out-competes the M strain(Figure7). The results show that reducing the waiting times decreases the time it takes for the B strain to out-compete the M strain (Figure8).”

Point-by-Point responses to Reviewer #2

This study tries to assess the feasibility of use of co-infecting virus to combat problematic virus, based on simulation results. The main conclusion obtained by the authors is that if co-infecting (harmless) virus can accumulate more rapidly than the target (harmful) virus and if the two viruses compete for the same type of host cells, the target virus will be dominated by the co-infecting virus, even when the co-infecting virus came later; to be honest, this conclusion will not surprise anyone, but I can find a potential value of this study in establishment of a framework for discussion on “under what circumstances such antiviral strategy can function.” To fulfill the potential value of the study, I recommend the authors to consider the following points:

 We thank the reviewer for her/his careful and critical reading of our manuscript and his/her valuable comments and suggestions. We revised and improved our manuscript accordingly. The responses to the comments are as follows. We revised and improved our manuscript accordingly.

MajorPoints:

C2-1) The simulation results are totally dependent on parameter values, especially on lambda. Please provide lambda (and other parameter) values used for the simulations explicitly in every figure; to avoid confusion, clearly define parameters. For example, define different lambda value for each virus by using lambda B and lambda M.

A2-1) The more explanation about parameters have been now included in the figures.

C2-2) Please provide average waiting times for each step, as readers can compare them with D.

A2-2) There are three different concepts related to time in our manuscript: Waiting time, simulation step and co-infection delay.

Waiting time is the duration of each stage of the viral growth process. Waiting times are generated from Exponential or Weibull distributions. The waiting times are scaled by common units of time such as milliseconds. For example, if the generated value for one waiting time is ten, we consider the time length of the corresponding stage as ten milliseconds. In the simulation model, different Gillespie algorithms—different viral growth processes—are run at the same time. Since, the time lengths of the algorithms are different, we assume the constant time length for the simulations step, such as L. One step of the simulation includes a number of Gillespie algorithms which the sum of their time lengths is L. The plot at the below shows the number of Gillespie algorithms(reactions) in 200 steps. The five stages of the viral growth process are simulated in each reaction. As the time length of each distinct stage is different,—generated from Exponential or Weibull distribution—there is no unique waiting time for each reaction. Therefore, it is not possible to provide the average for waiting times.

“Co-infection delay”(D) is the delay between the infection of the tissue by each strain. In other words, D is the number of steps at which the infection by the second strain starts. For example, D=30 means the second strain introduced to the model thirty steps after introduction of the first strain. Regards to the difference between the concepts of co-infection delay and waiting times, it is not possible to compare them.

C2-3) Figures and figure legends do not correspond to each other in some instances. I need explanation for Fig. 3 but I wonder where I can find it.

A2-3) We are sorry for the confusion we have caused. The figure numbers are now edited.

C2-4) In Fig. 6, B virus continues to accumulate but M virus does not. Could the authors provide intuitive explanation of this phenomenon? For example, limitation in the number of host cells will not explain because B virus would also suffer.

A2-4) The number of virions is not accumulated during the simulation. They are counted in each step, separately. To make more clarification, we have included some explanation about counting the virions and infected cells in the Method. The text reads:

“Infection calculation: Number of Virions and infected cells

The number of infected cells and free virions are calculated in each step of the simulation. The time length of each step is predefined before starting the simulation. Hence, the time length of the steps is constant. In our study, the time length of the simulation steps is about 1000 milliseconds. Co-Wish runs a number of Gillespie algorithms simultaneously. In other words, many cells can be infected at the same time.The time lengths of the algorithms with different settings are not alike. Each step is terminated when the sum of the time length of different runs of Gillespie algorithm is equal to the determined time length of the step.”

In our study, virus B—benign virus—has a shorter growth process than virus M. This property allows the virus to spread and reproduce faster than its competitor. Therefore, virus B can occupy more host cells. In other hands, when one cell is infected the cells in its vicinity are at risk to be infected by the same virus. Due to there is no assumption about host cell division, Virus B is a winner virus in the competition between two viruses and does not let virus M to infect the host cells. As it is obvious in Figure 6 in all assumed conditions, referred to different co-infection delay values for virus B, the slope of the plots are steeper for virus B. 

C2-5) In relation to the above comment, I’m not sure if the authors assumed (1) limitation in the number of cells, (2) host cell division, and (3) innate immunity in each figure.

A2-5) More explanation about the simulated tissue has now been included in the “The dynamics of single strain infection” part of the revised version of the manuscript. The lines [138]-[141] of the revision now reads:

 “To visualize the viral infection of tissue, Co-Wish simulates the tissue as a lattice in which each node represents one cell. The dimension of the lattice is determined by the user. In our simulations, the host cell division is not designed. In addition, there is no competition between host cells in Co-Wish. ”

The status of the immune system has now been included in the figure legends (Figures 2,4,6,7,8). In addition, the explanation about simulated immune system has now been revised:

“In Co-Wish, the immune system response is characterized by the probability at which the immune system can eliminate the toxic elements (γ), the delay between the infection and the immune response, and the capacity of the immune system (I). Toxic elements include one or both of the following: infected cells; virions. In other words, Co-Wish can eliminate the infected cells, the virions, as well as the combination of both. Here, the capacity refers to the maximum number of each kind of toxic element that can be eliminated by the immune system. The capacity value has to be less than the number of toxic elements that are potentially produced in the early steps of the simulation. γ is drawn from a uniform distribution, U(a; b), that its parameters are within [0; 1]. The immune system can eliminate toxic elements with a different probability. This characteristic indicates that all toxic elements produced in each step of the simulation do not have the same probability of elimination. For example, Co-Wish generates separate probability values for the killing of each virion. Here, we assume that a virion is killed with probability p and survives with probability 1 − p. ” 

C2-6) In my understanding, authors think that competition for host cells is important in the simulation. Could the authors provide the time-course change of the number (and/or proportion) of cells attacked by both of the virus (before they get infected)? I’m not sure if the local spread of viruses assumed in this manuscript allows competition for host cells, though it totally depends on the virus and host cell densities assumed.

A2-6) The competition between two strains is over the host cells of the simulated tissue. The winner strain is the one which occupies more number of the cells than its competitor. In the proposed simulation model, the dimension of the lattice representing a tissue is determined before starting the simulation. The host division and competition between host cells is not assumed. Thus, the number of safe cells (but are potential to be infected by one of the two strains) is decreasing during the simulations. If one strain infects a cell, the cells in the vicinity of the infected cell are at risk to be infected by the strain. Thus, the rival strain has a poor possibility to infect those cells. That is how the competition between strains is constructed. The spread of infection follows a designed pattern. Inspired by the dispersion principle in physics, we designed the spread of the infection around one infected cell. The part “the spread of the viral infection” in Method describes the pattern. In addition, as it is mentioned in the (A2-5) we have now included more explanation about the simulated tissue to the revised version of the manuscript.

C2-7) L40 and L312: the authors touch incubation times of viruses but it was not clear to me how they are relevant to this study.

A2-7) The proposed simulation model does not consider the incubation times. Talking about the different incubation time of viruses in the introduction part is just to explain how “competitive exclusion principle” could be applied about co-infection of a tissue by different viral strains. To make more clarification about the simulation assumptions, we have now altered the explanation about the assumptions of our model in the introduction part. The introduction part in the revision, lines [11]-[41] now reads:

“The crux of this work is to couple the CEP with the life cycle of viruses. To model how two types of viruses, B and M, co-infect a tissue, we have to make a few assumptions: 

1. Different viruses can have eclipse phases of varying lengths. The period between the initial infection and the first detectable viremia is known as the eclipse period—from the moment a virus enters the cell until it starts assembling its progenies and subsequences burst out of the cell. The duration of this period varies among different virus strains and affects the kinetic of infection (e.g., the eclipse period of SHIV lasts around a day [3] while it lasts between 7 to 8 hours in SARS-CoV [4].

2. Different viruses have different burst sizes, i.e., different per-cell virion particles. This assumption is reasonable, but even for SARS-CoV-2, the estimated burst size of 103 is simply based on MHV-2 data (e.g., [5]). The variation in particle-to-PFU among animal viruses [6] can be used to deduce the veracity of this assumption, but more direct data on SARS-CoV-2 is needed.

3. The CEP applies when two virus strains compete to infect a cell. In a spatially heterogeneous environment, different populations tend to partition the environment into non-overlapping micro-environments; for one of the most famous experimental demonstrations of such a situation see [9]. However, the displacement of one of the competitors by another is inescapable when the competitors cannot adapt or construct new niches in the environment in a reasonable timescale. The notion of CEP is undisputed when it comes to animals and bacteria trying to occupy the same ecological niche, but its application to viral confection is not as unequivocal as one would imagine.

In addition, different viruses can have different incubation times. For example, the median incubation period—i.e., the period between the onset of infection and the appearance of symptoms—for SARS-CoV-2 is estimated to be 5.1 days [7] (although such estimates should be taken with a grain of salt, e.g., [8]), whereas the incubation period for the common cold is around 1 − 3 days, and it could even be as long a few months to few years (e.g., Rabies and AIDS) [6]. Here, we do not consider any designation to simulate the different incubation times of two imaginary viruses.” 

 Other minor points:

C2-8) Figure 2 Legend: Replication -> replication

A2-8) It is now edited.

C2-9) L178: ith -> with

A2-9) It is now corrected.

C2-10) The reference formatting is terrible.

A2-10) It is now edited. 

Point-by-Point responses to Reviewer #3

In this paper the authors have presented an interesting framework for the progression of viral infection in host tissue. The model simulates the invasion of two competing viruses in a theoretical lattice of initially susceptible cells. Cells that are infected by one virus are prevented by being simultaneously infected by a competing strain to simulate the effect of the competitive exclusion principle (CEP). The premise of this paper therefore may be appealing to not only mathematical modellers but also biologists and medics working in related fields.

However, I do have some key concerns, that if addressed would benefit the manuscript:

Thank you for the reviewer encouraging comments. We also thank you for your detailed review of our work and for your valuable and pertinent suggestions. We have edited the manuscript along the lines suggested by you. We believe that the presentation of our research has now been improved considerably. The responses to the comments are as follows.

C3-1) The authors do not provide a strong justification for the application of their model to SARS-CoV-2 in their introduction, which detracts from the narrative since the reader is left questioning the relevance of the model in this setting. Some justification is provided later in the discussion section, however, this is not particularly strong, and thus the application of the model to SARS-CoV-2 feels very extrapolative. I would therefore encourage the authors to consider removing the emphasis of the model’s application to SARS-CoV-2 in the abstract and introduction of the paper.

A3-1) The abstract and the introduction sections are now revised according to the reviewer ‘s suggestion. In the abstract of the revision text now reads:

“Inspired by the competition exclusion principle, this work aims at providing a computational framework to explore the theoretical feasibility of viral co-infection as a possible strategy to reduce the spread of a fatal strain in a population. We propose a stochastic-based model—called Co-Wish—to understand how competition between two viruses over a shared niche can affect the spread of each virus in infected tissue. 

To demonstrate the co-infection of two viruses, we first simulate the characteristics of two virus growth processes separately. Then, we examine their interactions until one can dominate the other. We use Co-Wish to explore how the model varies as the parameters of each virus growth process change when two viruses infect the host simultaneously. We will also investigate the effect of the delayed initiation of each infection. Moreover, Co-Wish not only examines the co-infection at the cell level but also includes the innate immune response during viral infection.

The results highlight that the waiting times in the five stages of the viral infection of a cell in the model—namely attachment, penetration, eclipse, replication, and release—play an essential role in the competition between the two viruses. Finally, we discuss how co-infection with some forms of common respiratory diseases, such as common cold, might be useful in tackling a viral respiratory pandemic, such as the current COVID-19 crisis.”

At the beginning of the introduction:

“The “competitive exclusion principle” (CEP), also—perhaps erroneously—known as Gause’s law" [1], is the consequence of natural selection operating on non-interbreeding populations that occupy the same ecological niche. As Darwin put it, “ the competition will generally be most severe [...] between the forms which are most like each other in all respects” [2] (p.320). In the simplest reading, this principle implies that, in the competition between two sympatric non-interbreeding populations over the same ecological niche, one will displace the other.”

C3-2) The action of the CEP in the model was buried in the paper (it was briefly alluded to in lines 201 -203). This could have been made much clearer. Moreover, it appears that the authors are more specifically modelling superinfection exclusion, whereby secondary infection is prevented at a cellular level, which I would consider to be a specific example of the CEP.

A3-2) We agree with the reviewer that we should have put more emphasis on the CEP in our manuscripts. The introduction has been revised accordingly (as explained in A3-1). In particular, we highlighted the CEP and its application in modeling co-infection at a cellular level by addressing it at the beginning of the introduction. 

C3-3) The paper would benefit if the model was described in clearer detail. For example, the authors have not made the action of the immune response in the model clear. I appreciate that to appeal to a wider audience that the authors may not wish to go into a great level of detail in the main body of the text, but I feel a more detailed description of all of the components of the model in the supplementary information would be very beneficial.

A3-3) The simulated immune system has been described in the “The influence of the immune system” part of the results section. In the revised manuscript, the explanations have now been edited and completed due to your comments (C3-10,C3-11,C3-12).

In addition, we have prepared a manual for the designed simulator. The manual file includes the description of all the components of the simulation model. It is now at:

https://github.com/safarvafadar/virusinfection

Parts of the manual explanation about immune system arguments are as follows:

Immune System Properties of the immune system

 Description Activation of the immune system is optional. The designed immune system could resemble as innate response to the viral infection.

Arguments: 

Killing probability A positive value from [0,1] to determine the probability at which immune system eliminates the virions or infected cells.

Delay A positive integer to specify the step at which the immune system activates.

Killing Mode The designed immune system is able to eliminate the infected cells (Whole-cell) or free virions (Part of the population) or a combination of both (Mix). 

Capacity There are two options: Limited and Unlimited. If Limited is selected, a positive integer indicates the maximum number of infected items which will be eliminated by the immune system. The immune system will be deactivated after approaching the capacity. By selecting Unlimited, the immune system will be active to the end of the simulation.

C3-4) The paper would benefit from a more thorough justification of the parameterisation of the model, with suitable references to existing literature. While I appreciate different parameter combinations have been explored, this is somewhat limited, and I feel a more detailed sensitivity analysis would be beneficial. This also would most likely be appropriate to include in the supplementary information.

A3-4) Thanks for the suggestion. The results of simulations based on different values for the parameters plus more explanation about choosing the range of the parameters of the model have been now included in the supplementary file , part A. The plot at the below shows the results of different simulations based on ten values for the parameter λ for generating waiting times at replication stage:

To determine the range of the parameters value we ran different simulations, for a single virus model, by changing the parameter values and made different combinations of the different parameter values. Waiting times, the times that each virus spends in each stage of viral infection, were chosen from Exponential or Weibull distribution. To shorten the time length of one stage, we should increase the parameter λ of the Exponential distribution (λ>1) and decrease the shape parameter of the Weibull distribution(ω<1). After running different simulations, we figured out that shortening the time length of the viral infection process affects the number of infected cells (and produced virions) but the total trend is similar in all the simulations. Therefore, we decided to report just the simulation in which the parameter λ is 3 and the parameter ω is 0.5.

 C3-5) The authors do not appear to use time units in their parametrisation of the model, or in the results presented, where all figures are presented in “steps” along the x-axis.

A3-5) The X-axis in all the figures represents the number of simulation steps (200 steps). We did not use the time units such as “second”. The complete explanation about the reason for using a number of simulation steps as timescale are in A3-6. Please, read the explanation at A3-6.

C3-6) The above point raises significant concerns – since the model is based on a Gillespie algorithm, each time step will be of a different time length due to different sojourn times between events. It could therefore be misleading to plot the trajectories of their results (e.g. number of virions) against the step count for each run of the simulation, since these are evolving over different timescales. I would therefore strongly advice that the model is revisited to ensure that all x-axes are presented over some units of time (hours or days).

A3-6) As you mentioned, the time length of each Gillespie algorithm to simulate the viral infection process is different. This characteristic of Gillespie algorithm is the main reason that we have displayed the results over the number of steps. In other words, in our simulations different Gillespie algorithms are run in each infected cell simultaneously. Since the time lengths of the different algorithms are not alike, it is difficult to calculate the total number of new infected cells (or virions) in each unit of time. To solve the problem, we decided to define the steps of the simulation with constant time length. In our study, the time length of the simulation steps is about 1000 milliseconds. One step is terminated when the sum of the time length of different Gillespie algorithms runs equal to the determined time length of the step. This designation gives us some suitable advantages:

- Providing the clearer view about the simulation process.

- Providing the possibility of comparing different simulations, since the results of all the simulations displayed over the same unit of time.)

- Providing the possibility of calculating the average number of infected cells and virions in each step. 

The plots at the below show the time lengths and the number of Gillespie runs of 200 steps of one simulation. In the stepSum plot, there are slight differences between height of the bars because for some steps the sum of the time lengths of the Gillespie algorithms does not exactly equal to the determined time length for the step.

More explanations about the applied Gillespie algorithm have now been included in the manuscript. The introduction of the revision, lines [83]- [92] now reads:

“To simulate the stochastic multi-stage nature of the viral growth process occurring in viral co-infection, Gillespie algorithm, also known as the stochastic simulation algorithm, is usually considered as a trustworthy alternative. In particular, it supports simulating both discrete and stochastic behaviors of each stage of viral growth. The algorithm explicitly simulates the behavior of each virus in each stage of its growth process. Finally, it can provide a flexible framework to study the stochastic effects induced during the evolution of the infection. One characteristic of our simulation model is that several Gillespie algorithms run in infected cells, simultaneously. This characteristic can represent what we presume of viral infection in one tissue. Gillespie’s algorithms are indeed independent of their features.” 

In the Method lines[438]-[446]:

“- Infection calculation: Number of Virions and infected cells

The number of infected cells and free virions are calculated in each step of the simulation. The time length of each step is predefined before starting the simulation. Hence, the time length of the steps is constant. In our study, the time length of the simulation steps is about 1000 milliseconds. Co-Wish runs a number of Gillespie algorithms simultaneously. In other words, many cells can be infected at the same time.The time lengths of the algorithms with different settings are not alike. Each step is terminated when the sum of the time length of different runs of Gillespie algorithm is equal to the determined time length of the step.”

C3-7) Additionally, a thorough spelling and grammar check is required.

A3-7) The manuscript has now been thoroughly edited.

While I do have a significant number of concerns with the manuscript in its current form, I do believe that a valuable piece of work could result from the model framework that the authors have used. Moreover, I felt the premise of the model is interesting in principle and would like to thank the authors for sharing their work.

I have made some additional inline comments below:

C3-8) line 69 - 70: If there is evidence of a reduced risk of a subsequent coronavirus infection by another respiratory virus then I think this needs to be explicitly stated to be able to back up the paper's justification for applying the model to SARS-CoV-2. Currently, it's not specified which pairs of viruses had a negative interaction between them in ref [10].

A3-8) More explanation about the viruses with negative interactions have been now included in the manuscript, introduction section lines [50]-[61]:

“There are two major lines of investigations that can illuminate the applicability of the CEP to viral infections in general, and SARS-CoV-2 in particular. These include (i) the experimental evidence on the negative interactions between viruses, and (ii) the computational models of co-infection. In relation to the first line of research, a recent paper [10] employed a population-level approach, based on 44230 cases over 9 years, to track the epidemiological interactions between 11 strains of respiratory viruses, including influenza A and B, rhinoviruses, and three human coronaviruses (229E, NL63, HKU1). The authors have shown both positive and negative interactions between pairs of viruses over time They inferred negative interaction between Rhinoviruses (A-C) and Influenza A virus—at both population and host levels—and negative interaction between Influenza A virus and Influenza B virus—at the population level. In addition, the three human coronaviruses showed negative interactions with Rhinoviruses (A-C), human Respiroviruses 1 and 4 at the population level.”

3-9) Line 133: The legend for figure 2 does not appear to match figure 2 in this document, which appears to show the different spreading patterns (square vs cross)

A3-9) Thank the reviewer for his careful consideration. It is now corrected.

C3-10) line 165: Does "the capacity of the immune system (I)" have any units, such as the maximum number of infected cells/virions eliminated per day? It is not clear how "I" is defined mathematically in the model.

A3-10) The parameter “I” is the maximum number of virions (or infected cells) that would be eliminated by the immune system in each simulation step. It is a positive integer and its value is chosen empirically. Determining the capacity depends on the number of virions (or infected cells) which are produced in the early steps of the simulations, the dimensionality of the lattice. The capacity value should be chosen less than the number of virions which potentially can be produced at early steps of the simulation. Thus, in each step some virions would be allowed to be survived. In our simulations, “I” is set as 100 because during different simulations we figured out in each step the number of produced virions is more than 100.

It is also worth mentioning that in the Co-Wish simulator user is able to set different values for parameter I. More explanation about setting the simulation parameters such as “I” are included in the prepared manual of the simulator.

In addition, more explanation about the parameter “I” has been included in the manuscript line[158] – [166]. The text now reads:

 “ In Co-Wish, the immune system response is characterized by the probability at which the immune system can eliminate the toxic elements (γ), the delay between the infection and the immune response, and the capacity of the immune system (I). Toxic elements include one or both of the following: infected cells; virions. In other words, Co-Wish can eliminate the infected cells, the virions, as well as the combination of both. Here, the capacity refers to the maximum number of each kind of toxic element that can be eliminated by the immune system. The capacity value has to be less than the number of toxic elements that are potentially produced in the early steps of the simulation. ” 

C3-11) line 167: Gamma appears to have been defined in lines 163-164 as "the rate at which the immune (system) eliminates infected cells or virions". Therefore, it presumably has units of "cells/virions cleared per hour/day". I therefore do not understand the logic behind why it is then drawn from a U [0,1] distribution.

A3-11) The simulated immune system eliminates each virion (or infected cell) with different probability. In other words, all the virions produced in each step of the simulation do not have the same probability to be eliminated. The simulator generates a separate probability value for each virion (or infected cell). The value “ p ” for the parameter γ means the virion will be eliminated with probability “ p ” and it will be survived with probability “1-p”.

In the revised manuscript, we have replaced the word ‘rate’ with the word ‘probability’. In addition, we have now included more explanation about the parameter γ to the ‘The influence of the immune system’ part of the manuscript:

“In Co-Wish, the immune system response is characterized by the probability at which the immune system can eliminate the toxic elements (γ), the delay between the infection and the immune response, and the capacity of the immune system (I). Toxic elements include one or both of the following: infected cells; virions. In other words, Co-Wish can eliminate the infected cells, the virions, as well as the combination of both. Here, the capacity refers to the maximum number of each kind of toxic element that can be eliminated by the immune system. The capacity value has to be less than the number of toxic elements that are potentially produced in the early steps of the simulation. γ is drawn from a uniform distribution, U(a; b), that its parameters are within [0; 1]. The immune system can eliminate toxic elements with a different probability. This characteristic indicates that all toxic elements produced in each step of the simulation do not have the same probability of elimination. For example, Co-Wish generates separate probability values for the killing of each virion. Here, we assume that a virion is killed with probability p and survives with probability 1 − p. ” 

C3-12) Line 176: to differentiate this from the "unlimited immune response capacity" situation, I think it would be better to have notation along the lines of 0.

A3-12) We have modified the notations. In the part of “The influence of the immune system” of the revision text now reads:

“ 1. Weak with limited capacity (WL): γ ∼ U(0; 0:4)

 2. Weak with unlimited capacity (WU): γ ∼ U (0; 0:4)

 3. Strong with limited capacity (SL) : γ ∼ U(0:6; 1)

 4. Strong with unlimited capacity (SU) : γ ∼ U(0:6; 1)

Limited capacity indicates that the immune system will not be active in all the steps of the simulation. It will be deactivated when the capacity hits the predetermined value for I. If the capacity is unlimited, the immune system will be active by the end of simulation.” 

C3-13) line 199: It is unclear how the second virus is seeded into the model. It would be helpful if the authors clarified this (e.g. is it seeded into a cell in the lattice at random?)

A3-13) The location of the second virus is chosen randomly. We have now included more explanation about locating the second virus to the “Competition during co-infection” part. The text now reads:

“The location of the second virus in the lattice is chosen randomly, and it is independent of the first virus location. Those cells that are not infected by the first virus can be a niche of the second virus.”

C3-14) line 213: There is a lack of justification for the parameters chosen for the various distributions in table 1. While there may be limited literature for SARS-CoV-2, literature on other respiratory viruses may allow the authors to justify their choices. As highlighted before, I feel the paper would benefit if all times are given in appropriate units such as days, rather than being treated as dimensionless. I would also encourage the authors to consider including a sensitivity analysis of the different parameters in the supplementary information.

A3-14) As we have mentioned in (A3-4) more explanation about choosing the parameters value and the results of more simulations with different values of the parameters have been now included in the supplementary files. To design the viral infection process, we reviewed different literature about different viruses. Reviewing the studies informed us about the limitations of deterministic mathematical methods such as ODE. Therefore, we decided to propose a stochastic approach to study the viral (co-)infection. A comparison between deterministic and stochastic models are in the introduction part,lines[75]-[89].

 The arguments of the model were determined based on the different studies about viruses including SARS-CoV-2 and other respiratory viruses. For example, we figured out that the eclipse stage, among different stages of viral infection, has a strong influence on the process. Therefore, to generate the waiting times of the eclipse phase we applied Weibull distribution considering the shape of its hazard function. Using Weibull distribution gives us the ability to control the hazard rate of the event of transition to the next stage. The explanation about the reason for applying Weibull distribution is in the main text lines [368] – [384].

As it is explained in the (A3-6) we displayed the results of the simulations over the number of steps. We hope our explanations in A3-6 could clarify the reasons for using a number of steps as timescale.

C3-15) lines 267-271: Hepatitis G is now typically referred to as GB virus C.

A3-15) It is now corrected.

C3-16) lines 312-314: If the authors wish to keep the focus of the paper on SARS-CoV-2, I feel the manuscript would benefit if these studies (refs [30] and [31]) were highlighted in the introduction section. The reader would then understand the authors justification for the model to be placed in the context of SARS-CoV-2.

A3-16) Following the reviewer’s suggestions in C3-1 and C3-2, the manuscript, the abstract, and introduction parts have now been revised. We believe that the presentation of our research has now been improved considerably.

---

## [Decision Letter · Decision Letter 1]

9 Dec 2020

PONE-D-20-22300R1

Competitive exclusion during co-infection as a strategy to prevent the spread of a virus: a computational perspective

PLOS ONE

Dear Dr. Sadeghi,

Thank you for submitting your manuscript to PLOS ONE. After careful consideration, we feel that it has merit but does not fully meet PLOS ONE’s publication criteria as it currently stands. Therefore, we invite you to submit a revised version of the manuscript that addresses the points raised during the review process.

We look forward to receiving your revised manuscript.

Kind regards,

Bashar Ibrahim

Academic Editor

PLOS ONE

Reviewers' comments:

Reviewer's Responses to Questions

**Comments to the Author**

1. If the authors have adequately addressed your comments raised in a previous round of review and you feel that this manuscript is now acceptable for publication, you may indicate that here to bypass the “Comments to the Author” section, enter your conflict of interest statement in the “Confidential to Editor” section, and submit your "Accept" recommendation.

Reviewer #1: All comments have been addressed

Reviewer #3: All comments have been addressed

2. Is the manuscript technically sound, and do the data support the conclusions?

Reviewer #1: Yes

Reviewer #3: Partly

3. Has the statistical analysis been performed appropriately and rigorously? 

Reviewer #1: Yes

Reviewer #3: N/A

4. Have the authors made all data underlying the findings in their manuscript fully available?

Reviewer #1: Yes

Reviewer #3: Yes

5. Is the manuscript presented in an intelligible fashion and written in standard English?

Reviewer #1: Yes

Reviewer #3: Yes

6. Review Comments to the Author

Reviewer #1: The corrections regarding my revision were done well. I would agree to the publication of the paper now.

Reviewer #3: I would like to thank the reviewers for addressing the comments myself and the other reviewers had made in response to the manuscript's initial submission, and believe the presentation of the work has improved as a result. I do nevertheless have some ongoing concerns:

1) While I am thankful that the authors have given an improved explanation behind their choice of parameters in the supplementary information, I still believe that the paper would benefit significantly if these were placed in the context of some real world viruses. Currently the choice of parameters, both in the main body of the text and the supplementary information, are entirely theoretical and have not been justified. I feel this would be acceptable if the paper was presented as a wholly theoretical piece of work, however, the authors ultimately conclude that "Co-Wish suggests that common respiratory viral infections - such as common human coronaviruses - may limit the replication of SARS-CoV-2" (lines 317-318). Yet, when the choice of parameters is entirely theoretical, I do not believe such conclusions should be drawn, or even alluded to.

2) The results of the paper are still presented over steps rather than units of time. While I appreciate the authors responded to this point when I raised it previously, I still feel that it would be important to translate the results to be presented over time units if conclusions such as the one highlighted above are to be drawn. Moreover, it should not be too difficult for the authors to record the sojourn times between events from a Gillespie algorithm.

While I still have these concerns, I do hope that the authors are not too disheartened and want to stress that I still believe it is an interesting piece of work. I do however believe that the authors should consider taking one of either of the following approaches:

- Revisiting their model to present their results in units of time, with a choice of parameters that have been justified in some way from common respiratory viruses. I appreciate that there will be a lot of unknowns for the waiting times between different stages for real world viruses, but I believe it would be possible to make at least sensible ballpark estimates. If this were to be done, then I feel the authors would have more justification to make conclusions that could be placed in a real world context.

- However, if the authors do not wish to re-run their model and results, as suggested above, I feel the narrative of the paper should be revisited to emphasize that it is an entirely theoretical piece of work, and that any conclusions in relation to any real world viruses should not be made or alluded to.

7. PLOS authors have the option to publish the peer review history of their article (what does this mean?). If published, this will include your full peer review and any attached files.

Reviewer #1: No

Reviewer #3: **Yes: **Andrew Glover

---

## [Author Response · Author response to Decision Letter 1]

9 Jan 2021

Reviewer #3: I would like to thank the reviewers for addressing the comments myself and the other reviewers had made in response to the manuscript's initial submission, and believe the presentation of the work has improved as a result. I do nevertheless have some ongoing concerns:

We would like to thank the reviewer’s suggestions and encouraging words. The responses to his comments are as follows: 

C1) While we are thankful that the authors have given an improved explanation behind their choice of parameters in the supplementary information, we still believe that the paper would benefit significantly if these were placed in the context of some real world viruses. Currently the choice of parameters, both in the main body of the text and the supplementary information, are entirely theoretical and have not been justified. I feel this would be acceptable if the paper was presented as a wholly theoretical piece of work, however, the authors ultimately conclude that "Co-Wish suggests that common respiratory viral infections - such as common human coronaviruses - may limit the replication of SARS-CoV-2" (lines 317-318). Yet, when the choice of parameters is entirely theoretical, I do not believe such conclusions should be drawn, or even alluded to.

A1) 

We followed your suggestion; accordingly, we have edited the manuscript to emphasize the theoretical aspects of the study. 

In the Abstract of the revision, the text reads:

“While it could prove challenging to fully understand the therapeutic potentials of viral co-infection, we discuss that our theoretical framework hints at an intriguing research direction in applying co-infection dynamics in controlling any viral outbreak’s speed.”

In the Introduction of the revision, the paragraph below is now omitted:

“Based on the above assumptions, we propose that the competitive exclusion principle can, at least theoretically, be useful in tackling a viral pandemic, such as the current COVID-19 crisis, since one viral strain might be able to out-compete and eliminate a rival strain in the same ecological niche. Considering the current pandemic, this question offers hope as a means to control the novel coronavirus outbreak by supplanting the SARS-CoV-2 virus by a much less fatal one.” 

The Discussion section has been edited as well; for example, some explanations about the SARS-CoV-2 virus is now deleted. 

● At the beginning of the discussion, the following text is now dropped:

“We looked into the dynamics of viral co-infection to investigate the possibility of eliminating malignant virus strains in the population by pitting it against a more benign strain.”

The text lines [230]-[233] now reads:

“In this study, we look into the dynamics of viral co-infection using a CEP-inspired computational model. To explore the possible effects of co-infection by two virus species, we develop a stochastic-based viral infection simulator—called Co-Wish—to model and display the competition between two strains.”

● In the discussion, the paragraph below is now omitted:

“Following the same line of thought, we examined the possibility of slowing down the spread of COVID-19 through the introduction of a more harmless virus to a population. Our results offer the possible mitigating role of benign respiratory virus infection as a way of antiviral treatment in COVID-19 individuals. Accordingly, there is a clear and pressing need to conduct clinical trials to assess the role of co-infection in the clinical outcomes of SARS-CoV-2 infected patients.”

● Revised discussion focuses more on the theoretical aspects of the proposed models. We only discuss that our theoretical framework hints at an intriguing research direction in the dynamics of co-infection. Accordingly, the fifth and sixth paragraphs in the previous version are summarized into one paragraph—highlighted in green in the revised manuscript—which only concisely and suggestively discusses the possibility of controlling the spread of any viral infection, such as the SARS-CoV-2 virus, in the population by supplanting the lethal virus with a much less fatal one.

C2) The results of the paper are still presented over steps rather than units of time. While I appreciate the authors responded to this point when I raised it previously, I still feel that it would be important to translate the results to be presented over time units if conclusions such as the one highlighted above are to be drawn. Moreover, it should not be too difficult for the authors to record the sojourn times between events from a Gillespie algorithm.

A2)

Regarding the definition of one step in our simulation, we present the results over almost equal-length segments of assumed time. One step is terminated when the sum of the time length of different runs of the Gillespie algorithm equal to, at least, the determined time length of steps. In our simulation, the length of steps is about 1000msec; we represent the results once every 1000msec.

The summary of our reasons for displaying the results by using step number are as follows:

- Avoiding Computational Costs:

Using Discrete-Time Simulation (DTS), for example, displaying the results at each millisecond, imposes a lot of computational overhead such as CPU,IO, and RAM overhead. The computational costs make the simulation slow. 

Instead, we apply the TIME ADVANCE MECHANISM (TAM) approach and a parallel timeline for each infected cell viral growth process.

- Preparing Concise view about the parallel simulations: 

We believe that presenting the results over a number of steps can illustrate a more concise view about the parallel simulations held in different virtual infected cells than showing them over time. The main reason for this decision is simulations at infected cells are in parallel with different lengths of time. Showing the results over the number of steps allows the viral growth process to get finished in more infected cells. Figure 1 displays the main idea of the simulation. 

Figure 2 shows the simulation results over time unit (millisecond) just for one run. In our study, each simulation includes one hundred runs. The trajectory is very dense because it represents a large number of points. It seems it does not convey more information than displaying the results over the number of steps. 

Figure 1- schematic picture of viral infection simulation. Simulations in “n” cells are simultaneous. Different clocks illustrate different time lengths of the simulations. “Evaluate Time to Infect Neighbors” checks out the time length of the simulation. 

Figure 2- simulation results over time unit (millisecond) just for one run

---

## [Editor Report · Decision Letter 2]

13 Jan 2021

PONE-D-20-22300R2

Competitive exclusion during co-infection as a strategy to prevent the spread of a virus: a computational perspective

PLOS ONE

Dear Dr. Sadeghi,

Thank you for submitting your manuscript to PLOS ONE. After careful consideration, we feel that it has merit but does not fully meet PLOS ONE’s publication criteria as it currently stands. Therefore, we invite you to submit a minor revised version of the manuscript that addresses the points raised during the review process.

"Please elaborate more about the difference between your approach and other literature theoretical approaches particularly there is a huge number recent models and also software tools for SARS-CoV-2 and some on a related study. This would help and guide readers more in the introduction or discussion. I listed a few examples and more is even better in this case:   

10.20944/preprints202005.0376.v1 

10.1155/2020/4923856

10.1007/s00018-019-03382-0

https://doi.org/10.1016/j.bihy.2008.05.003

10.3390/v13010014

"

We look forward to receiving your revised manuscript.

Kind regards,

Bashar Ibrahim

Academic Editor

PLOS ONE

---

## [Author Response · Author response to Decision Letter 2]

31 Jan 2021

Dear Professor Bashar Ibrahim

Competitive exclusion during co-infection as a strategy to prevent the spread of a virus: a computational perspective

We are thankful for your consideration of our manuscript and for the constructive point you mentioned. We have attached the revised version of our manuscript (with modifications tracked in blue). 

This version of the manuscript has followed the suggestion we have received from the editor. Accordingly, we have discussed and analyzed the proposed model with other theoretical approaches. The third version of our manuscript now includes two dedicated paragraphs explaining the differences between the proposed method and other available theoretical approaches. These new paragraphs have been incorporated into the discussion section of the revised version. 

We hope that the revised manuscript could address the comment pointed out by you. 

We would like to thank you and the reviewers again for the careful reading of our work and the excellent comments we have received.

Mehdi Sadeghi 

Associate Prof.

National institute of genetic engineering and biotechnology

Tehran 

Iran

---

## [Editor Report · Decision Letter 3]

3 Feb 2021

Competitive exclusion during co-infection as a strategy to prevent the spread of a virus: a computational perspective

PONE-D-20-22300R3

Dear Dr. Sadeghi,

We’re pleased to inform you that your manuscript has been judged scientifically suitable for publication and will be formally accepted for publication once it meets all outstanding technical requirements.

Kind regards,

Bashar Ibrahim

Academic Editor

PLOS ONE
---

## [Editor Report · Acceptance letter]

11 Feb 2021

PONE-D-20-22300R3 

Competitive exclusion during co-infection as a strategy to prevent the spread of a virus: a computational perspective 

Dear Dr. Sadeghi:

I'm pleased to inform you that your manuscript has been deemed suitable for publication in PLOS ONE. Congratulations! Your manuscript is now with our production department. 

Kind regards, 

on behalf of

Dr. Bashar Ibrahim 

Academic Editor

PLOS ONE